# Watermark Robustness and Radioactivity May Be at Odds in Federated Learning

## Abstract

Federated learning (FL) enables fine-tuning large language models (LLMs) across distributed data sources. As these sources increasingly include LLM-generated text, provenance tracking becomes essential for accountability and transparency. We adapt LLM watermarking for data provenance in FL where a subset of clients compute local updates on watermarked data, and the server averages all updates into the global LLM. In this setup, watermarks are *radioactive*: the watermark signal remains detectable after fine-tuning with high confidence. The $p$-value can reach $10^{-24}$ even when as little as $6.6\%$ of data is watermarked. However, the server can act as an *active adversary* that wants to preserve model utility while evading provenance tracking. Our observation is that updates induced by watermarked synthetic data appear as outliers relative to non-watermark updates. Our adversary thus applies strong robust aggregation that can filter these outliers, together with the watermark signal. All evaluated radioactive watermarks are *not robust* against such an active filtering server. Our work suggests fundamental trade-offs between radioactivity, robustness, and utility.

## 1 Introduction

Large language models (LLMs) are increasingly used to generate synthetic datasets for fine-tuning due to the high cost of collecting human annotations and expert knowledge (Taori et al., 2023; Wang et al., 2023; Li et al., 2023). Such synthetic datasets often coexist with sensitive natural data in domains such as healthcare (Nik et al., 2023; Nikolentzos et al., 2023) and finance (Harsha et al., 2025). As a result, privacy concerns and regulations such as GDPR (Voigt & Von dem Bussche, 2017) and HIPPA (U.S. Congress, 1996) restrict the direct sharing of such synthetic, sensitive data.

Federated learning (FL) has emerged as a framework for collaboratively fine-tuning LLMs across distributed data sources (Kairouz et al., 2021; Rieke et al., 2020; Tian et al., 2022; Caldas et al., 2019). In FL, clients train models locally and only share model updates with a central server that aggregates them into a global model. While this setup helps with privacy by reducing data exposure, it does not address data provenance, i.e., attributing data contributions to their providers. Recent regulations recognize provenance as essential for accountability and transparency in AI (European Parliament and Council of the European Union, 2024), yet its implications in FL remain largely unexplored.

A promising direction toward provenance is watermarking the synthetic data generated from LLMs by embedding secret signals that can be statistically detected. Prior work shows that models fine-tuned on watermarked LLM-generated text exhibit *radioactivity* in a centralized setting, where the watermark signals remain detectable after fine-tuning (Sablayrolles et al., 2020; Sander et al., 2024) However, watermarking in FL introduces new challenges. Training for several epochs on local data together with the non-IID nature of client data introduces noise and causes client drift (Shi et al., 2022). These can reduce the watermark signal below a statistically detectable level. Therefore, it is unclear whether existing watermarks are radioactive in FL.

Moreover, the server in FL may act as an *active adversary*, deliberately attempting to evade provenance tracking. This is a new threat model for watermark *robustness* in FL. In non-FL setups, prior work shows that continued fine-tuning on non-watermarked (clean) data can substantially reduce watermark detectability (Sander et al., 2024). However, in FL the server does not control the fraction of watermarking clients and does not have access to clean data. At the same time, the server

must maintain the global LLM utility, so it cannot arbitrarily remove clean updates. This raises the important question: Does there exist an effective attack to remove watermarks in FL without sacrificing model utility?

In this paper, we show that existing watermarks are radioactive in FL but active adversaries can remove these. The key observation is that there is a clear separation between updates from clean clients and watermarking ones. We illustrate this in Figure 1 where we use t-SNE (Maaten & Hinton, 2008) to visualize the high-dimensional model updates of clean and watermarked data. The problem of watermark removal reduces to robust aggregation in FL (Diakonikolas et al., 2018; Choudhary et al., 2024; Lee et al., 2025). We thus propose to use a filtering algorithm that removes model updates outside the variance of the distribution of clean updates, i.e., as outliers of the distribution[1]. We show that none of the evaluated radioactive watermarks are robust against such filtering algorithms.

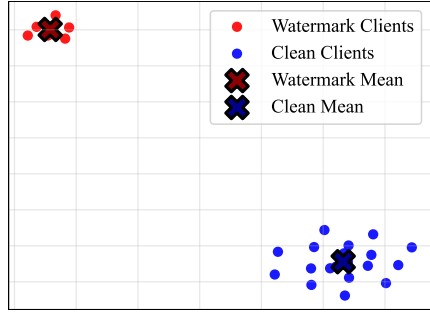

Figure 1: t-SNE visualization of model updates shows the clear differences in updates from clean clients (blue) and watermark clients (red).

To the best of our knowledge, this is the first work to introduce federated data provenance (Section 3). In summary, our contributions are the following:

- We first adapt existing watermarking schemes to vanilla (benign) federated LLM fine-tuning settings where the server averages client updates (Reddi et al., 2021). We empirically demonstrate that watermarked data is *radioactive in FL* such that watermarked LLM-generated data is detected with high statistical significance, with $p$-values ranging from $10^{-3}$ to $10^{-24}$.

- We further formulate the active adversary threat model. We realize it through state-of-the-art robust aggregators that filter watermarked updates. The active adversary successfully removes the watermark on *all evaluated setups* that were radioactive under the vanilla setting.

- We provide an extensive evaluation showing that none of the current watermarks achieve *radioactivity, robustness* and *utility* at the same time under our evaluated setups.

Our findings open future research directions into understanding the fundamental limitations of watermarking and designing better schemes for guaranteeing data provenance in FL.

## 2 PRELIMINARIES

### 2.1 FEDERATED SETUP

In the federated fine-tuning setup, we consider a system of $N$ clients $C = \{c_1, c_2, \ldots, c_N\}$ that collaboratively train a global LLM model $\mathcal{M}_\theta$ under the coordination of a central server. The LLM has model parameters represented as a vector $\theta \in \mathbb{R}^d$, where $d$ denotes the number of parameters. Each client $c_i$ maintains a private dataset $D_i$ stored locally. At the beginning of each communication round $t$, the server distributes the current global model parameters $\theta^t$ to all clients. Each client then fine-tunes on $D_i$ and produces a local update, $\Delta\theta_i^t$. The server collects and aggregates all these updates via an aggregation function $\text{AGG} : \mathbb{R}^{N \times d} \to \mathbb{R}^d$ to update the global model for the next round: $\theta^{t+1} = \theta^t + \text{AGG}(\{\Delta\theta_i^t\}_{i=1}^N)$. Let $\mathcal{M}_\theta^t$ and $\mathcal{M}_\theta^{t+1}$ represent the global model at the start and end of round $t$, respectively. Let $\mathcal{M}_\theta$ be the final model after training.

### 2.2 LLM WATERMARKING

Let $\mathcal{M}_\omega$ be an LLM that takes as input a sequence of tokens (prompt) $\pi = (x_1, \ldots, x_q) \in \mathcal{V}^q$ and generates a probability distribution $p \in [0,1]^{|\mathcal{V}|}$, where $\mathcal{V}$ is the vocabulary of the model. It then samples the next token from this probability distribution using a procedure such as top-k sampling (Fan et al., 2018; Radford et al., 2019) or greedy decoding (Germann et al., 2001). This process repeats autoregressively to generate an output sequence $\mathbf{x} \in |\mathcal{V}|^Q$, denoted as $\mathcal{M}_\omega(\pi) \to \mathbf{x}$.

---

[1]More analysis of updates across watermark methods and data are included in Appendix F.

To watermark the outputs of $\mathcal{M}_\omega$, a generation function $\text{WATERMARK}_s^{\mathcal{M}_\omega}(\pi) \to \mathbf{x}^w$ employs a secret key $s$ to perturb the decoding process of $\mathcal{M}_\omega(\pi)$. The perturbation inserts a detectable watermark signal and produces a watermarked response $\mathbf{x}^w$ (Zhao et al., 2025). The function $\text{DETECT}_s(\mathbf{x}) \to \{\textit{True}, \textit{False}\}$ performs a statistical test on $\mathbf{x}$ to detect whether $\mathbf{x}$ is produced by $\mathcal{M}_\omega$. This function takes the secret key $s$ as input and returns *True* if $\mathbf{x}$ contains a watermark signal consistent with $s$, and *False* otherwise. Different watermarking schemes achieve this perturbation in distinct ways. At each generation step, KGW+ (Kirchenbauer et al., 2024) hashes the previous $k$ tokens and $s$ to generate a pseudo-random subset of $\mathcal{V}$, termed the *green list*. The watermark perturbs the decoding by biasing the sampling to favor the tokens from the *green list*. KTH+ (Kuditipudi et al., 2024) does not rely on hashing. Instead, it pre-defines a random number sequence with $s$ and embeds the sequence to sampling in a way that preserves the output distribution of $\mathcal{M}_\omega$.

**Radioactivity.** Let $\mathcal{M}_\omega$ be the model that generates a watermarked dataset $D^w$. To evaluate whether another model $\mathcal{M}_\theta$ has fine-tuned on watermarked data, a modified detection function $\text{DETECT}_s^{\mathcal{M}_\theta}(D^w)$ is used. Instead of operating on $D^w$, this variant examines the radioactivity of $\mathcal{M}_\theta$'s prediction on $D^w$ (Definition 1, (Sander et al., 2024)). Specifically, $\text{DETECT}_s^{\mathcal{M}_\theta}(D^w)$ first computes an accumulated score over $\mathcal{M}_\theta$'s predictions on $D^w$. It then performs a statistical test $T$ by comparing this observed score to the null distribution, i.e., the distribution of scores on the output of $\mathcal{M}_\theta$ which was not trained on $D^w$. The resulting $p$-value indicates the probability of the observed score occurring by chance. We compare it with a predefined significance level to output a binary decision: *True* or *False*. For more details on how KGW+ and KTH+ accumulate score and compute the null distribution, see Appendix A.

**Definition 1 (Radioactivity)** *Dataset $D^w$ is $\alpha$-radioactive for a statistical test $T$ with $H_0$: Model $\mathcal{M}_\theta$ was not trained on $D^w$, if the test $T$ can reject $H_0$ at a p-value below the significance level $\alpha$.*

## 3 FEDERATED DATA PROVENANCE

We study data provenance in FL via watermarking. In FL provenance, an $\epsilon$ fraction of clients, that we denote as watermarking clients, aim to prove that their datasets were used to train the global model $\mathcal{M}_\theta$. Using the same watermark generation algorithm $\text{WATERMARK}_s^{\mathcal{M}_\omega}$, these clients watermark their local dataset which results in watermarked dataset $D_i^w$. At round $t$, all clients send local updates $\Delta\theta_i^t$ and the server aggregates them, as presented in Section 2.1. We denote all local updates sent to the server at round $t$ as $U_\Delta = \{\Delta\theta_i^t\}_{i=1}^N$. We denote the updates computed using the watermarked dataset as $W_\Delta$ and the non-watermarked (clean) as $C_\Delta$. Watermarking clients can verify their contribution using $\text{DETECT}_s^{\mathcal{M}_\theta}$.

We consider two settings depending on the role of the server in FL. In the vanilla setup (**VanillaFL**), the server averages the updates from all clients $\mathcal{T}(U_\Delta, \theta^t) = \theta^t + \text{AVG}(\{\Delta\theta_i^t\}_{i=1}^N) = \theta^t + \frac{1}{N}\sum_{i=1}^N \Delta\theta_i^t$ (Reddi et al., 2021). The challenge in VanillaFL is whether the watermark remains detectable after training, as the clients' local model updates can drift from the global model and dilute the watermark signal. This drift occurs because the updates are sent to the server after multiple local epochs on non-IID data. Thus, VanillaFL serves as a baseline for radioactivity in FL. However, the server lacks incentive to participate in the watermarking scheme. We refer to this as the *active adversary* setup (**ActiveFL**). Its goal is to obtain a global model that evades detection of the watermark while maintaining the LLM utility. We describe the threat model below.

**Threat Model.** It is assumed that there are no privacy attacks where other clients or the server may try to infer information about each client's local training data. In ActiveFL, the server can only change the aggregation and must follow the rest of the FL protocol. In VanillaFL, the server follows the FL protocol, averaging updates. All client data are kept private from the server. All clients are honest and follow the FL protocol. For simplicity, we assume watermarking clients have a shared key $s$ and generative model $\mathcal{M}_\omega$. Watermarking clients share a key $s$ and a generative model $\mathcal{M}_\omega$ used for generated synthetic data, which are unknown to the server. More details are in Appendix B.

### 3.1 PROBLEM STATEMENT

In this paper, we consider federated data provenance under the active adversary $\mathcal{A}$ threat model. Our adversary $\mathcal{A}$ takes as input the updates $U_\Delta$, where $|C_\Delta| = 1 - \epsilon > 0.5$, and the current global

model parameters $\theta^t$ to return the updated model parameters $\theta^{t+1}$. The adversary in ActiveFL aims to ① obtain an updated model $\mathcal{M}_\theta^{t+1}$ that has a similar utility to $\mathcal{T}$ on a set of clean updates under some evaluation metric $\mathcal{E}^2$ and ② reduce the detectability of the watermark. Specifically, it aims to reduce the statistical significance $\alpha$ of a given watermarked dataset $D_i^w$ for $\mathcal{M}_\theta^{t+1}$. We use $\text{DETECT}_s^{\mathcal{M}_\theta^{t+1},\mathcal{T}}(D_i^w)$ to denote that the detection test is run on the predictions of $\mathcal{M}_\theta^{t+1}$ on the watermarked dataset $D_i^w$ at round $t$, where $\mathcal{M}_\theta^{t+1}$ is obtained by training with $\mathcal{T}$. We use $\approx_\mathcal{E}$ to denote similar utility under the metric $\mathcal{E}$. Note that radioactivity (Definition 1) is defined for a given dataset $D^w$, not a watermarking scheme (Section 2.2). We thus also define *robustness* with respect to $\mathcal{A}$ and to a dataset that is $\alpha$-radioactive had it been updated with $\mathcal{T}$.

**Definition 2 (FL Robustness)** *Let $D_i^w$ be an $\alpha$-radioactive dataset for a statistical test $T$ and the model $\mathcal{M}_\theta^{t+1}$ obtained in VanillaFL with $\mathcal{T}$ such that $\text{DETECT}_s^{\mathcal{M}_\theta^{t+1},\mathcal{T}}(D_i^w) \to$ True at round $t$. If there exists an adversary $\mathcal{A}$ such that for every round $t_\mathcal{A}$: ① $\mathcal{A}(U_\Delta, \theta^{t_\mathcal{A}}) \approx_\mathcal{E} \mathcal{T}(C_\Delta, \theta^{t_\mathcal{A}})$ and ② $\text{DETECT}_s^{\mathcal{M}_\theta^{t_\mathcal{A}+1},\mathcal{A}}(D_i^w) \to$ False, then $D_i^w$ is not robust to $\mathcal{A}$.*

The robustness definition is *counterfactual*, similar to its counterpart for watermarked generations of an LLM (Zhao et al., 2025). It has a hypothetical precondition that at some round $t$, the dataset $D_i^w$ would become radioactive on the resulting global model $\mathcal{M}_\theta^{t+1}$, if it had been updated with $\mathcal{T}$. Given that $D_i^w$ satisfies this precondition, if the server runs $\mathcal{A}$ for each round $t_\mathcal{A}$ and the dataset $D_i^w$ is not radioactive on the resulting model $\mathcal{M}_\theta^{t_\mathcal{A}+1}$, then we say that the $D_i^w$ is not robust to $\mathcal{A}$. On the contrary, a watermarked dataset $D_i^w$ is robust to $\mathcal{A}$ if it remains detectable in spite of $\mathcal{A}$ updating the model for all rounds $t_\mathcal{A}$, $\mathcal{A}(U_\Delta, \theta^{t_\mathcal{A}})$. While increasing the rounds $t$ means that the dataset's radioactivity increases (lower $p$-value), it comes at a cost of overfitting which can damage the final model utility. If the datasets are not radioactive to start with, then the server trivially satisfies condition ②. If the server does not satisfy condition ①, then condition ② is easy to satisfy. For instance, adding noise to all updates could sufficiently perturb the watermark signal but such a strategy can deteriorate the final $\mathcal{M}_\theta$ utility. Note that the clients and server have conflicting goals regarding watermark detection, yet all parties aim to maintain model utility. Our paper addresses whether there exists such an active server that satisfies Definition 2.

## 4 APPROACH

Figure 2 presents an overview of ActiveFL, where we propose an active server $\mathcal{A}$ that removes the watermarked updates by filtering the $W_\Delta$ from $U_\Delta$ at each round $t$. The challenge is in distinguishing watermarked updates from clean ones as the server must preserve enough $\Delta\theta_i^t \in C_\Delta$ for effective learning. Our insight is that LLM watermarks like KGW+ are radioactive but low-distortion, i.e., there is a small statistical distance between the watermarked and clean data distributions (Zhao et al., 2025). If distortions from watermarked LLM-generated text propagate to updates computed on such texts, then there is a *measurable shift* in the watermarked vs. clean updates distributions. Specifically, we observe that updates $\Delta\theta_i^t \in W_\Delta$ become outliers in the distribution of clean updates $C_\Delta$. By removing outliers

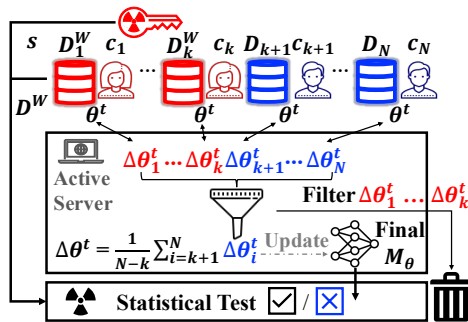

Figure 2: Overview of federated data provenance in ActiveFL for LLMs.

from client updates before averaging, the server can thus satisfy goals ① and ② simultaneously. Therefore, we employ strong Byzantine-robust aggregators that are designed to filter out outlier updates in our ActiveFL setup.

Byzantine robust aggregation algorithms estimate the true mean of model updates from clean clients while mitigating influence from corrupted ones (Huber, 1964). The aggregator FIL takes a set $U_\Delta$

---

[2]Evaluation metrics include log-likelihood on a test set or qualitative metrics such as BLEU (Papineni et al., 2002) or ROUGE (Lin, 2004).

of $N$ vectors in $\mathbb{R}^d$ (local updates from $N$ clients), where an $\epsilon$-fraction are arbitrarily corrupted as inputs. For the subset of uncorrupted (clean) vectors $C_\Delta \subseteq U_\Delta$, the aggregator FIL guarantees:

$$\|\text{FIL}(U_\Delta) - \mu_C\|_2 \leq \beta \cdot \|\Sigma_C\|_2^{\frac{1}{2}}$$

where $\mu_C = \frac{1}{|C_\Delta|} \sum_{\Delta\theta_i^t \in C_\Delta} \Delta\theta_i^t$ is the mean of clean vectors. This bound guarantees that the *bias*, the distance between the aggregator's output and the true mean, is limited by a multiplicative factor $\beta$ times the square root of $\|\Sigma_C\|_2$. Here, $\|\Sigma_C\|_2$ is the spectral norm of the covariance matrix of $C_\Delta$, representing the maximum variance of the uncorrupted vectors. Strong robust aggregators guarantee that $\beta$ is $O(1)$, independent of vector dimension $d$ (Diakonikolas et al., 2018). For details on FIL implementations, see Appendix C.

Algorithm 1 outlines our framework for integrating data provenance into FL fine-tuning. Our FL framework builds upon FedOpt (Reddi et al., 2021), incorporating two key modifications. First, to accelerate convergence, we broadcast the global model to all clients in each round $t$ (line 7), rather than to a subset as in FedOpt. In parallel, each client then performs local training on its own dataset using CLIENTOPT with learning rate $\eta_c$ and sends the resulting model updates to the server. The second adaptation is the aggregation step. The central server aggregates the received updates using a function AGG (line 11). FedOpt is a specialized case where AGG is simply averaging, which we adopt as VanillaFL. In ActiveFL, AGG is instead a strong Byzantine robust aggregation function. The server then updates the global model by applying SERVEROPT with learning rate $\eta_s$ on the aggregated results. The fine-tuning process uses early stopping, i.e., it terminates training when the evaluation loss ceases to decrease. Upon completion, the clients perform watermark detection on the final global model $\mathcal{M}_\theta^t$ using their own watermarked datasets.

---

**Algorithm 1** LLM Watermarks in Federated Learning Finetuning

---

    **Input** Clients $C$, local datasets $\{D_i\}_{i=1}^N$, initial global model parameters $\theta^0$, server learning rate $\eta_s$, client learning rate $\eta_c$, local training steps $J$

1: **for** each client $i \in \mathcal{C}$ **do**
2:     **if** client $i$ chooses to apply watermark **then**
3:         $D_i^w = \text{WATERMARK}_s^{\mathcal{M}_\omega}(D_i)$
4:         $D_i \leftarrow D_i^w$                       $\triangleright$ Client $i$ uses $D_i^w$ as local dataset
5: Initialize $t = 0$
6: **while** validation loss decreases **do**                    $\triangleright$ Early stopping
7:     $\theta_{i,0}^t = \theta^t$              $\triangleright$ Global model broadcast parameters to all clients
8:     **for** each client $i \in \mathcal{C}$ **in parallel do**
9:         Train on $D_i$ with CLIENTOPT and $\eta_c$ for $J$ steps
10:         $\Delta\theta_i^t = \theta_{i,J}^t - \theta^t$                 $\triangleright$ Compute local updates
11:     $\Delta\theta^t = \text{AGG}(\{\Delta\theta_i^t\}_{i=1}^N)$
12:     $\theta^{t+1} = \text{SERVEROPT}(\theta^t, -\Delta\theta^t, \eta_s)$         $\triangleright$ Update global model
13:     $t \leftarrow t + 1$
14: Each client $i$ runs $\text{DETECT}_s^{\mathcal{M}_\theta^t}(D_i^w)$

---

A distributional shift typically exists between synthetic and natural datasets, even without watermarking applied. We find that the distributional shift also contributes to FIL effectively removing $W_\Delta$. We therefore conduct an ablation study where all clients fine-tune with synthetic dataset to further examine watermark robustness in Section 5.4.

## 5 EVALUATION

In this section, we ask the following research questions:

**(RQ1)** Are LLM watermarks radioactive in the federated learning settings?
**(RQ2)** Are LLM watermarks robust against strong Byzantine robust aggregation?
**(RQ3)** How do watermark hyperparameters affect the trade-off between watermark radioactivity and robustness?
**(RQ4)** Can watermark radioactivity and robustness be enhanced simultaneously? If so, does this introduce a broader three-way (radioactivity, robustness and utility) trade-off?

Table 1: LLM watermarks radioactivity under FL fine-tuning with $\epsilon = 6.6\%$. Pre-fine-tuning $p$-value on $\mathcal{M}_\theta$ is $\sim 0.5$, consistent with $H_0$. KGW+ shows strong and model-size-dependent radioactivity. KTH+ shows no radioactivity.

| Data | WM | Model | $p$-value | |
|------|-----|-------|-----------|----------|
| | | | **Before FT** | **After FT** |
| C4 | KGW+ | 70M | 0.584 | 0.169 |
| | | 160M | 0.397 | $1.27 \times 10^{-3}$ |
| | | 410M | 0.877 | $2.41 \times 10^{-8}$ |
| | KTH+ | 70M | 0.485 | 0.500 |
| | | 160M | 0.500 | 0.500 |
| | | 410M | 0.480 | 0.480 |
| Alpaca | KGW+ | 70M | 0.204 | 0.013 |
| | | 160M | 0.309 | $1.59 \times 10^{-11}$ |
| | | 410M | 0.302 | $4.96 \times 10^{-24}$ |
| | KTH+ | 70M | 0.490 | 0.480 |
| | | 160M | 0.480 | 0.480 |
| | | 410M | 0.480 | 0.480 |

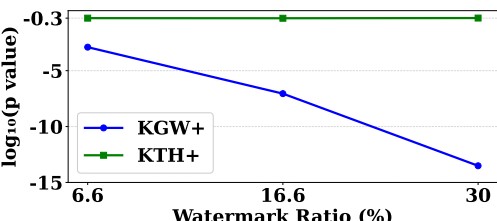

Figure 3: KGW+ radioactivity improves with larger $\epsilon$, while KTH+ remains not radioactive.

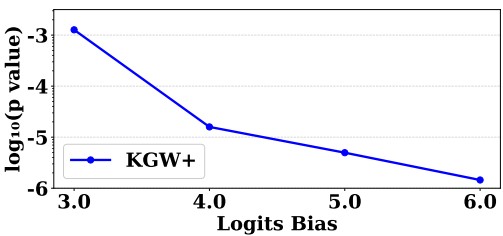

Figure 4: KGW+ radioactivity improves with larger $\delta$.

## 5.1 SETUP

We begin by briefly outlining the FL setup, LLM watermarking schemes, and evaluation metrics. For more details on experimental settings, see Appendix D.

**Models & Datasets.** We use Pythia (Biderman et al., 2023) as our global model and perform experiments on three different model sizes: 70M, 160M, and 410M. We evaluate the LLM watermarking schemes on two distinct datasets: C4 (raw text) (Raffel et al., 2023) and Alpaca (question-answer pairs) (Taori et al., 2023). Unless otherwise specified, all ablation studies use the Pythia-160M model and the C4 dataset. All experiments are performed on NVIDIA H100 GPU (80GB).

**Experimental Setup.** We present a total of 30 clients in our FL setup. Among these, $k$ clients apply the watermark to their local data, where $k \in \{2, 5, 9\}$ corresponds to a watermark ratio $\epsilon = \frac{k}{30} \in \{6.6\%, 16.6\%, 30.0\%\}$. ActiveFL uses RandEigen (Lee et al., 2025) as the strong robust aggregator. Fine-tuning stops after three consecutive rounds of worsening validation loss. We evaluate two representative LLM watermarking schemes: KGW+ (Kirchenbauer et al., 2024) and KTH+ (Kuditipudi et al., 2024). We employ the Pythia-2.8b model as the generative model. For the KGW+ baseline experiments, we use a softmax temperature ($T$) of 0.8 and a logit bias ($\delta$) of 3.0.

**Evaluation Metrics.** We examine global model utility using entropy loss. To evaluate watermark robustness, we introduce two additional metrics calculated in the first fine-tuning round. Let $L$ be the total number of layers. For each layer $\ell$, let $\mathcal{W}_\ell$ be the set of watermarking clients and $\mathcal{F}_\ell$ be the set of all clients filtered by the aggregator. Averaging across all $L$ layers, we define 1) **Evasion Rate (ER)** as the mean fraction of watermarking clients that remain after aggregation: $\mathrm{ER} = \frac{1}{L} \sum_{\ell=1}^{L} (1 - \frac{|\mathcal{W}_\ell \cap \mathcal{F}_\ell|}{|\mathcal{W}_\ell|})$; and 2) **Overfiltering Rate (OFR)** as the mean fraction of filtered clients that are not watermarked: $\mathrm{OFR} = \frac{1}{L} \sum_{\ell=1}^{L} (1 - \frac{|\mathcal{W}_\ell \cap \mathcal{F}_\ell|}{|\mathcal{F}_\ell|})$.

## 5.2 LLM WATERMARK RADIOACTIVITY IN FL

We first evaluate LLM watermark radioactivity by fine-tuning $\mathcal{M}_\theta$ under VanillaFL. We report $p$-value accumulated across all watermarked datasets for simplicity (see Appendix E.1 for details).

**Watermark Radioactivity.** KGW+ exhibits strong radioactivity. Even when the global model is fine-tuned on datasets containing only 6.6% watermarked samples, the KGW+ detection tests yield

Table 2: KGW+ robustness under FL with $\epsilon = 6.6\%$. KGW+ is not robust against RandEigen under ActiveFL.

| Data | Model | $p$-value | |
|------|-------|-----------|--------|
| | | **Vanilla** | **Active** |
| C4 | 160M | $1.27 \times 10^{-3}$ | 0.550 |
| | 410M | $2.41 \times 10^{-8}$ | 0.613 |
| Alpaca | 160M | $1.59 \times 10^{-11}$ | 0.231 |
| | 410M | $4.96 \times 10^{-24}$ | 0.282 |

Table 3: KGW+ robustness against RandEigen with varying watermark proportion ($\epsilon$). KGW+ robustness improves (higher ER) with a larger $\epsilon$. The KGW+ watermark is not robust for all evaluated $\epsilon$, with $p$-values around 0.5 after applying RandEigen.

| $\epsilon$ | $p$-value | | ER | OFR |
|------|-----------|--------|------|------|
| | **Vanilla** | **Active** | (%) | (%) |
| 6.6% | $1.27 \times 10^{-3}$ | 0.550 | 3.5 | 48.2 |
| 16.6% | $8.83 \times 10^{-8}$ | 0.517 | 8.4 | 53.6 |
| 30.0% | $3.10 \times 10^{-14}$ | 0.733 | 9.9 | 50.1 |

significantly low $p$-values. Table 1 shows that $p$-value can be as low as $10^{-8}$ on C4 and $10^{-24}$ on Alpaca. We expect that this strong radioactivity generalizes to other hashing-based watermarking methods (Fu et al., 2024; Lee et al., 2024; Aaronson & Kirchner, 2023), as they are all compatible with the accumulative detection mechanism. Unlike hashing-based methods, KTH+ is not radioactive in our FL setup. This is due to their weaker detection method. Specifically, their detector cannot accumulate statistical signal across prompts. Therefore, their watermark signal is not significant enough to be detected ($p$-value is around 0.5).

**Influence of $\mathcal{M}_\theta$ Size and $\epsilon$.** We observe that KGW+ radioactivity improves with the global model size. Table 1 shows that larger LLMs produce lower $p$-values for the same watermarked dataset. On dataset C4, increasing the model size from 70M to 410M parameters shifts the detection from ineffective to effective: we cannot reject $H_0$ for the 70M model ($p$-value $= 0.169 > 0.01$), but we can confidently reject $H_0$ for the 410M model ($p$-value $= 10^{-8} \ll 0.01$). Furthermore, Figure 3 shows that KGW+ radioactivity improves with larger $\epsilon$. Under identical settings of the baseline Pythia-160M experiments, the post-finetuning $p$-value drops sharply from $10^{-3}$ to $10^{-14}$ as $\epsilon$ increases from 6.6% to 30.0%. In contrast, KTH+ demonstrates weak radioactivity, regardless of model size or $\epsilon$. Figure 3 shows that $p$-value for KTH+ remains consistently around 0.5 across all evaluated settings. This again occurs because the detection signal cannot accumulate across prompts.

> **(RQ1)** Statistically distortion-free watermark (KTH+) is not radioactive in the FL setting, whereas KGW+ is radioactive. KGW+ radioactivity improves with larger $\epsilon$ and $\mathcal{M}_\theta$ size.

### 5.3 LLM WATERMARK ROBUSTNESS IN FL

We evaluate watermark robustness on configurations that are definitively radioactive (KGW+ on Pythia-160M and 410M) under VanillaFL. For these configurations, we compare the $p$-value after fine-tuning under ActiveFL (using the RandEigen aggregator) to those under VanillaFL.

**KGW+ Robustness.** Table 2 shows that KGW+ is not robust to the RandEigen aggregator. Fine-tuning with simple averaging yields significant post-finetuning $p$-values, all of which are smaller than or equal to $1.27 \times 10^{-3}$. In contrast, fine-tuning with RandEigen produces $p$-values that are statistically indistinguishable from random chance (around 0.5). This indicates that RandEigen effectively filters the watermark's signal during aggregation, preventing reliable detection in the final global model.

**Influence of $\epsilon$.** KGW+ watermark robustness improves with larger $\epsilon$. Table 3 shows that increasing $\epsilon$ from 6.6% to 30.0% raises the ER from 3.5% to 9.9%. Higher ER indicates that fewer watermarking clients are filtered out, thereby enhancing watermark robustness. This trend aligns with the theoretical limits of the RandEigen aggregator, which is only guaranteed to eliminate all gradient outliers when $\epsilon < 8.3\%$. However, despite the improvement in robustness, KGW+ remains undetectable (post-fine-tuning $p$-value $\geq 0.01$) for all evaluated $\epsilon$ values under ActiveFL. Therefore, none of the evaluated LLM watermarks is effective in the federated fine-tuning setting. The KTH+ watermark is not radioactive, while the KGW+ watermark is not robust to RandEigen aggregation.

Table 4: Filtering is less effective when all clients use synthetic data (ER $\geq$ 60.2%) which eliminates the shift introduced by $\mathcal{M}_\omega$. In all cases, the watermark is not radioactive.

| WM | Clean Client Dataset | ER (%) | OFR (%) |
|---|---|---|---|
| KGW+ | synthetic | 60.2 | 92.9 |
| | natural | 0.7 | 43.7 |
| KTH+ | synthetic | 60.7 | 92.9 |
| | natural | 0.2 | 46.2 |

Table 5: LLM watermark robustness with varying $\delta$. Higher $\delta$ makes filtering more effective (lower ER), increasing watermark robustness.

| WM | $\delta$ | $p$-value | | ER | OFR |
|---|---|---|---|---|---|
| | | Vanilla | Active | (%) | (%) |
| KGW+ | 0 | 0.268 | 0.201 | 60.2 | 92.9 |
| | 1 | $1.08 \times 10^{-5}$ | 0.370 | 21.6 | 80.6 |
| | 3 | $1.01 \times 10^{-18}$ | 0.274 | 1.0 | 46.2 |
| | 5 | $3.36 \times 10^{-12}$ | 0.788 | 0.7 | 40.7 |
| KTH+ | – | 0.480 | 0.480 | 60.7 | 92.9 |

> **(RQ2)** KGW+ watermark is not robust against strong Byzantine robust aggregation. KGW+ robustness improves with larger $\epsilon$ but remains undetectable even when $\epsilon$ reaches 30.0%.

## 5.4 WATERMARK HYPERPARAMETERS ANALYSIS

To evaluate how $\delta$ affects watermark radioactivity, we use the same setting as the baseline Pythia-160M experiment, varying only $\delta$. To evaluate watermark robustness, we test three hyperparameters: the generative model ($\mathcal{M}_\omega$), logits bias ($\delta$), and difference in softmax temperature used to generate clean and watermarked dataset ($\Delta T$). For each experiment, we vary one parameter while limiting the influence of the others. When limiting the influence of $\mathcal{M}_\omega$, clean clients fine-tune on synthetic datasets. To limit the influence of $\delta$ and $\Delta T$, we set them at 0.

**Influence of $\mathcal{M}_\omega$.** We quantify how the distribution shift between synthetic and natural data, which stems from the generative model ($\mathcal{M}_\omega$), undermines watermark robustness. When all clients fine-tune on synthetic datasets, the statistically distortion-free watermarks (KTH+) or low-distortion watermarks with $\delta = 0, \Delta T = 0$ (KGW+) introduce no distributional bias relative to the clean data. Table 4 shows that this alignment prevents the RandEigen aggregator from distinguishing clean clients from watermarking ones, resulting in high ER (60.2% for KGW+ and 60.7% for KTH+) and OFR (92.9% for KGW+ and KTH+). In contrast, when clean clients fine-tune on natural dataset, the aggregator effectively filters out watermarking clients (ER = 0.7% for KGW+ and 0.2% for KTH+). Recognizing that this distribution shift grants the server an advantage, we stress-test KGW+ robustness by eliminating the impact from $\mathcal{M}_\omega$ (i.e., having all clients fine-tune on synthetic data). The results demonstrate that KGW+ remains non-robust: at $\delta = 3$, its $p$-value increases from a significant $1.01 \times 10^{-18}$ under VanillaFL to a non-significant 0.274 under ActiveFL.

**Influence of $\delta$ on Radioactivity.** Consistent with prior analysis (Kirchenbauer et al., 2024), KGW+ radioactivity improves with a higher $\delta$. Figure 4 shows that $p$-value drops from $10^{-3}$ to $10^{-6}$ as $\delta$ increases from 3.0 to 6.0. However, this trend only holds within a specific range. An excessively low $\delta$ induces high repetition in the watermarked text. Since the detection method is designed to ignore such repetitions (Sander et al., 2024), the effective number of tokens evaluated becomes small. This compromises the statistical power of the detection test, leading to artificially low $p$-values under $H_0$ and a high false positive rate. Conversely, an excessively high $\delta$ makes the watermark pattern too random and complex for the global model to learn during fine-tuning, ultimately leading to high $p$-value after fine-tuning and a low true positive rate.

**Influence of $\delta$, $\Delta T$ on Robustness.** KGW+ robustness degrades with a higher $\delta$. Table 5 shows that both ER (decreasing from 60.2% to 0.7%) and OFR (decreasing from 92.9% to 40.7%) fall as $\delta$ increases from 0 to 5. Lower ER and OFR indicate that the RandEigen aggregator filters out watermarking clients more accurately, leading to worse watermark robustness. This is expected as a larger $\delta$ imposes a greater deviation from the clean data distribution. KTH+ is designed to be distortion-free, so its results are similar to those of KGW+ with $\delta = 0$. KGW+ robustness also degrades with a larger $\Delta T$: ER decreases from 60.2% to 1.8%, when $\Delta T$ increases from 0.0 to 0.8. For more details, refer to Table 8 in Appendix E.3.

> **(RQ3)** Distributional shift stemming from $\mathcal{M}_\omega$ reduces watermark robustness. Larger $\delta$ enhances KGW+ radioactivity but reduces robustness. Larger $\Delta T$ reduces KGW+ robustness.

## 5.5 UTILITY RESULTS

Although larger $\epsilon$ improves both watermark radioactivity and robustness, it degrades $\mathcal{M}_\theta$ performance. The entropy loss of the evaluation dataset increases from 3.156 to 3.161 as $\epsilon$ increases from 0% to 30.0%. Other evaluated benchmarks exhibit a similar trend, which is discussed in Appendix E. While $\epsilon$ has the potential to resolve the trade-off between radioactivity and robustness, it introduces a new, critical trade-off between watermark effectiveness and model performance. This places an upper bound on the practical value for $\epsilon$.

> **(RQ4)** Global model performance degrades with larger $\epsilon$. There is trade-off between model utility and watermark properties (radioactivity and robustness).

## 6 RELATED WORK

**LLM Watermarking.** LLM watermarking schemes fall into two main approaches: hashing-based schemes (Kirchenbauer et al., 2024; Aaronson & Kirchner, 2023; Christ et al., 2023) and non-hashing-based schemes (Kuditipudi et al., 2024; Zhao et al., 2023). Sander et al. (2024) shows that hashing-based watermarks exhibit radioactivity (KGW+ (Kirchenbauer et al., 2024)). We also consider KTH+ which is a non-hashing distortion-free scheme (Kuditipudi et al., 2024). Aaronson & Kirchner (2023) proposes another candidate watermark that could provide better radioactivity-robustness trade-off in FL. However, we find that it is not robust (Appendix G). Our work thus opens future research into understanding fundamental trade-offs and watermarking schemes in FL.

**FL Training.** Standard FL algorithms like FedSGD and FedAvg (McMahan et al., 2017) converge slowly under non-IID data (Karimireddy et al., 2021) or noisy environment (Zhang et al., 2020). To accelerate convergence, subsequent works employ adaptive local learning rate (Xie et al., 2020; Sun et al., 2023) and adaptive local interval (Spiridonoff et al., 2020; Ji et al., 2020). Our FL setup adapts the FedOpt framework (Reddi et al., 2021), which utilizes adaptive optimizers.

**Robust Aggregators.** Weak robust aggregators (Blanchard et al., 2017; Chen et al., 2019; Yin et al., 2018) compute dimension-wise centrality, suffering from a worst-case bias of $O(\sqrt{\epsilon d})$ (Lai et al., 2016; Lugosi & Mendelson, 2021). While polynomial-time strong robust aggregators (Diakonikolas et al., 2018; Hopkins et al., 2021; Kothari & Steurer, 2017) achieve dimension-independent bias bounds, they are often computationally expensive (Choudhary et al., 2024; Lee et al., 2025). We employ RandEigen (Lee et al., 2025) in our ActiveFL setup since it provides both strong bias bounds and quasi-linear runtime.

For further details on related work, including other LLM watermarks, post-hoc detection schemes, and backdoor approaches, refer to Appendix H.

## 7 CONCLUSION

We study the problem of federated data provenance for LLMs. We find that LLM watermarks are radioactive in FL, i.e., we can detect that watermarked synthetic data was used to train the global LLM with high confidence. We show a new threat model where the active adversary filters radioactive watermarks with strong robust aggregators. Our findings show that radioactivity and robustness to such adversaries are at odds in FL. We hope our work opens a new line of inquiry into understanding the fundamental limitations and designing better watermarking techniques for FL.

## 8 ETHICS STATEMENT

Our study investigates data provenance in federated learning using watermarks on synthetic LLM-generated text and does not involve interventions with human subjects or the collection of personally identifiable information. All experiments use public corpora C4 (Raffel et al., 2023) and instruction datasets Alpaca (Taori et al., 2023), plus synthetic generations produced offline for participating clients. We explicitly model an active server as adversary to surface risks such that robust aggregation can be misused to suppress provenance signals, which could undermine transparency and attribution. We therefore discuss this threat model, quantify detectability/robustness trade-offs, and avoid releasing any tool meant to remove third-party watermarks.

## 9 REPRODUCIBILITY

To ensure reproducibility, we will make the source code publicly available after acceptance. We specify the FL protocol and watermarking schemes. Our anonymized supplementary materials will include scripts to: (i) download/prepare datasets; (ii) generate watermarked text (KGW+, KTH+) with fixed seeds; (iii) run VanillaFL and ActiveFL; and (iv) compute all metrics from logged updates/summaries. We will also include exact hyperparameters, seeds, so others can reproduce numbers within expected stochastic variance.

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

## A  DETAILS ON LLM WATERMARKING

We adapt KGW+ and KTH+ to the radioactivity setting by applying the same cumulative scores (KGW+) or alignment-based statistics (KTH+) to model predictions on $D_w$, and we form nulls from models not trained on $D_w$. This section thus provides the generation and detection ingredients needed for those tests.

### A.1  (KIRCHENBAUER ET AL., 2024)

KGW+ is a hashing-based *green list* watermark. At step $t$, the scheme hashed the previous $k$ tokens together with a secret key $s$ to seed a PRNG that randomly partitions the vocabulary $V$ into a green list $G_t$ of size $\gamma|V|$ and a redlist $R_t$. The logits of green tokens are shifted by $\delta > 0$ before softmax. Sampling then proceeds from the biased distribution $\hat{p}^{(t)}$. Parameters $\gamma \in (0,1)$ and $\delta$ determine the trade-off of strength and quality of the watermark.

Given a tokenized text of $N$ tuples $X_N = \{x_i\}_{i=1}^N$, define the per-token indicator

$$\text{WScore}(x_i^{(0)}; s, x_i^{(-1)}, \dots, x_i^{(-k)}) \;=\; \mathbf{1}\{x_i^{(0)} \in G_i\}.$$

The cumulative score $S(X_N) = \sum_{i=1}^N \text{WScore}(\cdot)$ counts green tokens. Under $H_0$ where no watermark exists, the indicators are i.i.d. Bernoulli$(\gamma)$, in this case $S \sim \text{Binomial}(N, \gamma)$ and a one-sided $p$-value follows from the binomial CDF. A $z$-test form is

$$z = \frac{S - \gamma N}{\sqrt{N\gamma(1-\gamma)}},$$

with small one-sided $p$-value indicating detection. The watermark functioning by logit boost $\delta$ on $G_t$ preserves quality in low-entropy context, while still yielding predictable sensitivity curves as N grows.

## A.2 Kuditipudi et al. (2024)

KTH+ is a distortion-free, key-sequence watermark. A shared random key sequence $\xi = (\xi_1, \ldots, \xi_n)$ drives a decoder $\Gamma$ that maps $(\xi_i, p^{(t)})$ to a next token while preserving the model's original sampling distribution, marginalizing over $\xi$. Two instantiations are used:

- ITS (inverse-transform sampling): $\Gamma$ uses a uniform $u \in [0, 1]$ and a random permutation $\pi$ to pick the first token whose CDF which is ordered by $\pi$ exceeds $u$. This decoder is provably distortion-free.
- EXP (exponential-minimum, Gumbel-style): $\Gamma$ draws i.i.d. exponential variables keyed by $\xi$ and selects $\arg\min_i E_i/p_i$, yielding an equivalent distortion-free sample. In practice it also uses a shift-genenrate wrapper that slices fresh subsequences of $\xi$ to avoid reuse while retaining detectability.

The detection of KTH+ watermark is model- and prompt-agnostic. Given candidate text $y$ and shared key $\xi$, KTH+ aligns $y$ to $\xi$ using an alignment cost $d$ and then computes a nonparametric permutation test $p$-value by comparing the observed cost to costs under resampled keys $\{\xi^{(t)}\}$:

$$\hat{p} = \frac{1 + \sum_{t=1}^{T} \mathbf{1}\{\phi(y, \xi^{(t)}) \leq \phi(y, \xi)\}}{T + 1}, \quad \phi(y, \xi) = \min_{\text{blocks}} d(\text{block}(y), \text{block}(\xi)).$$

where $d$ is the alignment cost. A simple and fast alignment cost used in ITS is

$$d\big(y, (u, \pi)\big) = \sum_{i=1}^{\text{len}(y)} \big|u_i - \eta(\pi(y_i))\big|, \quad \eta(j) = \frac{j-1}{|V|-1}.$$

For edit robustness, KTH+ employs a Levenshtein-style variabt $d_\gamma$ that permits insertions or deletions with penalty $\gamma$. Under $H_0$ where key is independent of the text, the permutation test is valid and yields calibrated $p$-values. Statistical power grows exponentially in text length and only linearly degrades with key length.

## B  Threat Model

For simplicity, clients have local datasets of equal size, so no weighting across clients is required. If none of the clients apply watermarking, the data is independent and identically distributed. We have two types of clients. Watermarking clients are assumed to constitute $\epsilon$ of the total clients. All watermarking clients collude, sharing the same watermarking technique and random seed, without sharing their clean local data. Clean clients are local clients that compute updates using clean datasets without any watermarking. They operate independently, without communication or collaboration with watermark clients or among themselves. Each clean client has access only to its own local training data, which neither clients nor server cannot access. The central server is the active adversary. It controls the aggregation process of local updates, where the simplest method for aggregating is averaging (McMahan et al., 2017). The server is free to choose the aggregation function. Its goal is to obtain a model that evades detection of the watermark while maintaining the global model performance.

## C  Byzantine Robust Aggregation

We adopt the same notation from Section 4 for the Byzantine robust aggregation function. The aggregator takes a set $U_\Delta$ of $N$ vectors in $\mathbb{R}^d$, where an $\epsilon$-fraction are arbitrarily corrupted as inputs. For the subset of uncorrupted vectors $C_\Delta \subseteq U_\Delta$, the aggregators Fil guarantee:

$$\|\text{Fil}(U_\Delta) - \mu_C\|_2 \leq \beta \cdot \|\Sigma_C\|_2^{\frac{1}{2}}$$

where $\mu_C = \frac{1}{|C_\Delta|} \sum_{\Delta\theta_i^t \in C_\Delta}$, and $\|\Sigma_C\|_2$ is the spectral norm of the covariance matrix of $C_\Delta$. Robust aggregators security is closely related to the multiplicative factor $\beta$: a smaller $\beta$ corresponds

to a tighter defense. This factor distinguishes strongly-bounded aggregators from weakly-bounded ones. Weak robust aggregators have $\beta = O(d^{\frac{1}{2}})$, while strong robust aggregators have $\beta = O(1)$, independent of the vector dimension $d$ (Diakonikolas et al., 2018).

## C.1 POLYNOMIAL-TIME STRONG ROBUST AGGREGATORS

---

**Algorithm 2** Meta-Algorithm for Strong Byzantine Robust Aggregators

---

**Input** Watermark ratio $\epsilon$, partially watermarked updates $U_\Delta = \{\Delta\theta_1^t, \cdots, \Delta\theta_N^t\} \subseteq \mathbb{R}^d$, upper bound of clean covariance $\|\Sigma_C\|_2$ as $\Gamma$

**Output** Robust aggregated mean $\mu$

1: **for** $j = 0, \cdots, 2 \cdot \epsilon \cdot N - 1$ **do**
2:     Compute current maximum eigenvalue $\lambda_{curr} = \|Cov(U_\Delta)\|_2$
3:     **if** $\lambda_{curr} \le \Gamma$ **then** break
4:     **else**
5:         $U_\Delta \leftarrow$ OUTLIERREMOVALSUBROUTINE$(U_\Delta, \epsilon, \|\Sigma_C\|_2)$
6: **return** $\mu = \frac{1}{|U_\Delta|} \sum_{\Delta\theta_i^t \in U_\Delta} \Delta\theta_i^t$

---

Algorithm 2 shows a meta-algorithm for polynomial-time *strong* Byzantine robust aggregation functions. Polynomial-time strong robust aggregators (Diakonikolas et al., 2018; Hopkins et al., 2021; Kothari & Steurer, 2017) share a common strategy: they iteratively attenuate outliers in $X$ with an OUTLIERREMOVALSUBROUTINE until the bias is provably bounded. This subroutine, whose implementation varies by schemes, is designed to filter or down-weight outliers that contribute most to $X$ variance along the direction of largest eigenvectors. The iterative filtering continues until ① Number of iterations reaches $2\epsilon N$ (line 1), assuming at least one point is removed per round or ② largest eigenvalue of the current set's covariance matrix falls below a predefined threshold $\Gamma$ (line 3).

The first condition is derived from the proof that removing at least $2\epsilon N$ vectors is sufficient to eliminate all malicious vectors. The second condition originates from the observation that in practice the bias is bounded once the largest eigenvalue ($\lambda_{curr}$) falls below a predetermined threshold $\Gamma$, where $\Gamma = k\|\Sigma_Y\|_2$ for $k \in [\sqrt{20}, 9]$ (Zhu et al., 2023; Diakonikolas et al., 2019; Diakonikolas & Kane, 2019). Note that these methods require $O(\epsilon d^2)$ operations and $O(d^2)$ memory, making them impractical and vulnerable to attacks in high-dimensional settings (Choudhary et al., 2024; Lee et al., 2025).

## C.2 QUASI-LINEAR STRONG ROBUST AGGREGATORS

We employ RandEigen (Lee et al., 2025) in our ActiveFL. The strong Byzantine robust aggregator runs in quasi-linear running time, $O(Nd)$, and possesses provably near-optimal bias bounds. Unlike polynomial-time strong aggregators, it does not require computing the maximum variance of clean vectors, $\|\Sigma_C\|_2$. Instead, they replace the stopping condition in line 3 from Algorithm 2 with an equivalent heuristic based on eigenvalue convergence: the algorithm terminates once the maximum variance of the iteratively filtered set stabilizes. To avoid computational bottleneck, RandEigen also estimate the dominant eigenvectors through randomized dimensionality reduction rather than computing them exactly.

# D DETAILS ON SETUP

## D.1 FL SETUP

We adopt the FL framework from FedOPT (Reddi et al., 2021) to achieve faster convergence with good accuracy. Specifically, we employ ADAM (Kingma & Ba, 2017) as the server optimizer, which maintains optimizer states across communication rounds, and employ normalized SGD as the client optimizer. A constant learning rate of $10^{-5}$ is used for both optimizers. During each communication round, local fine-tuning is performed for a single epoch on the respective client datasets. We employ an early stopping strategy that terminates training once the evaluation loss on a clean, held-out validation split fails to decrease for three consecutive rounds.

Table 6: Per-Client Detection Results ($p$-values)

| Data | WM | Model | Client 1 | | Client 2 | |
|---|---|---|---|---|---|---|
| | | | **Before FT** | **After FT** | **Before FT** | **After FT** |
| C4 | KGW+ | 160M | 0.390 | $4 \times 10^{-3}$ | 0.447 | $4 \times 10^{-2}$ |
| | | 410M | 0.839 | $4.55 \times 10^{-5}$ | 0.725 | $5.84 \times 10^{-5}$ |

## D.2   LLM Watermarks Setup

We employ pretrained Pythia-2.8B as our generative model ($\mathcal{M}_\omega$) to ensure high-quality text generation and avoid potential biases that could arise from using the same models for $\mathcal{M}_\theta$ and $\mathcal{M}_\omega$ (Sander et al., 2024). On the question-answer pairs dataset Alpaca (Taori et al., 2023), we follow the original implementation to generate watermarked responses (Kirchenbauer et al., 2024). On the raw-text dataset C4 (Raffel et al., 2023), we treat the first 20 tokens as a prompt and generate the watermarked content to the same length as the original text. For the KGW+ baseline experiments, we use a softmax temperature of 0.8 and a logit bias of 3.0 following prior work's experimental setup (Sander et al., 2024). This configuration ensures watermark detectability without compromising text quality. For radioactivity detection, we adopt the setting of open model and supervised training data access, proposed by Sander et al. (2024)

## D.3   Dataset Setup

All clients have local datasets of the same size (i.e., the same number of prompts). In our baseline experiments, before any clients apply watermarks, we assume that all data are natural, non-watermarked, and IID across clients. When certain clients opt to watermark their dataset, they generate watermarked responses to all prompts from their original natural datasets, creating a watermarked, synthetic dataset. In the ablation study for watermark robustness, we examine the setup where the clean clients also use the same generative model to generate non-watermarked, synthetic dataset. The same evaluation conclusions hold for clean clients fine-tune on either natural or synthetic dataset.

To stress-test watermark robustness, we vary the decoding strategy for generating non-watermarked synthetic datasets based on the watermarking scheme.

1. **Kuditipudi et al. (2024)**: Since KTH+ does not employ a softmax temperature ($T$), we use standard Multinomial Sampling ($T = 1$) to generate non-watermarked dataset

2. **Kirchenbauer et al. (2024)**: KGW+ uses $T$ as a hyperparameter. We use a greedy decoding strategy ($T = 0$) to generate non-watermarked dataset. To create a softmax temperature difference ($\Delta T$) between watermarked and clean datasets, we set the watermarking scheme's $T$ to exactly $\Delta T$.

# E   Evaluation Results

## E.1   Individual Detection for Each Client

For simplicity, detection is run on the aggregated watermarked dataset (across all clients). However, this methodological choice does not affect our conclusions. To confirm, we examine per-client detection in the worst-case scenario: We use the C4 dataset with 6.6% KGW+ watermark, which is radioactive with aggregated detection but has relatively high $p$-values compared to other configurations. Table 6 shows that even in this worst-case scenario, per-client detection still yields high $p$-values before fine-tuning (around 0.5) and statistically significantly low $p$-value after fine-tuning ($p$-value $< 0.04$).

Table 7: Communication rounds before overfitting occurs for varied FL configurations.

| Data | WM | Model | Overfitting Round | |
|---|---|---|---|---|
| | | | Vanilla | Active |
| C4 | KGW+ | 70M | 31 | – |
| | | 160M | 37 | 35 |
| | | 410M | 33 | 30 |
| | KTH+ | 70M | 31 | – |
| | | 160M | 38 | – |
| | | 410M | 29 | – |
| Alpaca | KGW+ | 70M | 97 | – |
| | | 160M | 124 | 113 |
| | | 410M | 97 | 91 |
| | KTH+ | 70M | 104 | – |
| | | 160M | 136 | – |
| | | 410M | 99 | – |

Table 8: LLM watermarks robustness with varying $\Delta T$. Higher $\Delta T$ increases distributional shift, making filtering more effective (lower ER).

| WM | $\Delta T$ | ER(%) | OFR(%) |
|---|---|---|---|
| KGW+ | 0.0 | 60.2 | 92.9 |
| | 0.4 | 8.9 | 66.7 |
| | 0.8 | 1.8 | 44.3 |
| KTH+ | – | 60.7 | 92.9 |

### E.2 OVERFITTING POINTS

In Table 7, we report the exact number of communication rounds before overfitting occurs for the experiments presented in Table 1 and Table 2. We observe that dataset C4 needs the about 30 rounds to converge, while dataset Alpaca needs approximately 100 rounds to converge. Compared to VanillaFL, ActiveFL typically needs a few less rounds to converge.

### E.3 SOFTMAX TEMPERATURE DIFFERENCE

KGW+ robustness degrades with a higher $\Delta T$. A larger $\Delta T$ introduces a larger distributional shift, allowing RandEigen to filter out watermarking clients more effectively. Table 8 shows that increasing $\Delta T$ from 0.0 to 0.8 reduces ER from $60.2\%$ to $1.8\%$ and OFR from $92.9\%$ to $44.3\%$. $T$ is not a hyperparameter of KTH+. We only test the setting where the clean and watermark synthetic dataset share the same $T$ ($T = 1$ and $\Delta T = 0$), whose metrics are similar to those of KGW+ at $\Delta T = 0$.

### E.4 UTILITY OF THE GLOBAL MODEL

**Definition 3 (Utility)** *Let $\mathcal{M}_\theta(D^c)$ and $\mathcal{M}_\theta(D^w)$ be global models trained on a clean dataset $D^c$ and its watermarked version $D^w$, respectively. A watermarking scheme is $\lambda$-generalizable if, for any unseen dataset $D$ and any bounded evaluation metric $\mathcal{E}(\mathcal{M}, D) : \mathcal{M} \times D^* \to R$, the following holds:*

$$|\mathcal{E}(\mathcal{M}_\theta(D^c), D) - \mathcal{E}(\mathcal{M}_\theta(D^w), D)| \leq \lambda. \tag{1}$$

We evaluate the impact of watermark strength on downstream utility by varying the watermark ratio $\epsilon$ during training of 160M-Pythia, and assessing word-level perplexity on C4 (Raffel et al., 2023) and question-answering accuracy on MMLU-Pro (Wang et al., 2024b). As shown in Table 9, increasing $\epsilon$ from 0% to 16.6% slightly degrades both metrics, with higher perplexity and lower accuracy.

Table 9: Evaluation of 160M-Pythia with varying KGW+ watermark proportion ($\epsilon$).

| $\epsilon$ | Entropy Loss ↓ | C4 (Raffel et al., 2023) Word Perplexity ↓ | MMLU-Pro (Wang et al., 2024b) Accuracy ↑ |
|---|---|---|---|
| 0% | 3.1564 | 87.029 | 0.11760 |
| 6.6% | 3.1571 | 87.349 | 0.11735 |
| 16.6% | 3.1582 | 91.279 | 0.11652 |
| 30.0% | 3.1605 | 85.792 | 0.11827 |

However, at $\epsilon = 30\%$, the trend reverses, despite a marginally higher entropy loss of 3.1605, the model achieves its best C4 perplexity of 85.792 (vs. 87.029 at 0%) and highest MMLU-Pro accuracy of 0.11827 (vs. 0.11760).

A plausible explanation is that a sufficiently strong watermark acts as a data-dependent regularizer. The green list bias more consistently guides learning toward on-manifold continuations, implicitly smoothing labels by redistributing probability mass among plausible tokens and reducing gradient variance from rare tail tokens. When $\epsilon$ is too low (e.g., 6.6–16.6%), the bias perturbs the logits without providing these regularization benefits. At $\epsilon = 30\%$, however, the bias appears to surpass a useful-strength threshold, where improved calibration and stability outweigh the small increase in token-level cross-entropy, resulting in better generalization on both language modeling and reasoning tasks.

## F  LOCAL UPDATES ANALYSIS

From Figure 1 where clean clients trained on raw C4 data, we can observe clear and significant separation between the updates from clean clients and those from watermarking clients. This distinct separation strongly indicates that the watermarking data alters the model updates in a detectable manner, making the two groups of clients easily distinguishable in the high-dimensional space. We can also observe that the gradients from watermarking clients are similar to each other, forming a very tight and dense cluster. This highlights a high degree of local similarity and consistency among the watermark updates. This indicates that watermark's effect is consistent and pronounced and it could potentially be detected by using deviation from the robust mean estimation. Figures 5a and 5b visualize the impact of watermarking on client updates when clean clients use synthetic data. Figure 5a shows that watermark and clean updates still form two well-separated groups in high-dimensional space, though the two clusters are much closer than in Figure 1. In contrast, Figure 5b shows that the KTH+ watermark does not have obvious effect on client updates and clean and watermarking clients are not well separated in high-dimensional space. This result is consistent with the low radioactivity reported in Table 3.

For each update $\Delta\theta_i^1$ from clients in the first round, we measure the $\ell^2$-norm of the displacement from the clean mean, $\|\Delta\theta_i^1 - \mu_Y\|_2$. Figure 6 shows the distribution of these $\ell^2$-norms for KGW+ and KTH+ in the first round when clean clients use synthetic data. From Figure 6a, we observe that, even with synthetic data for clean clients, KGW+ tends to produce larger $\ell^2$-norms for watermarking clients. Thus, these updates can still be filtered out in the first round. By contrast, KTH+ induces little separation, and the distributions for clean and watermarking clients are very similar.

## G  OTHER WATERMARKS

### G.1  LLM WATERMARKS

Other than KGW+ (Kirchenbauer et al., 2024) and KTH+ (Kuditipudi et al., 2024), we examine another representative LLM watermark, Gumbel (Aaronson & Kirchner, 2023), which is computationally distortion-free, provided hashing context (previous $k$ tokens, $k$-gram) never repeats (Zhao et al., 2025). As a hashing-based method, Gumbel should exhibit radioactivity similar to KGW+ in the FL setup (Sander et al., 2024). However, its robustness hinges on achieving distortion-freeness

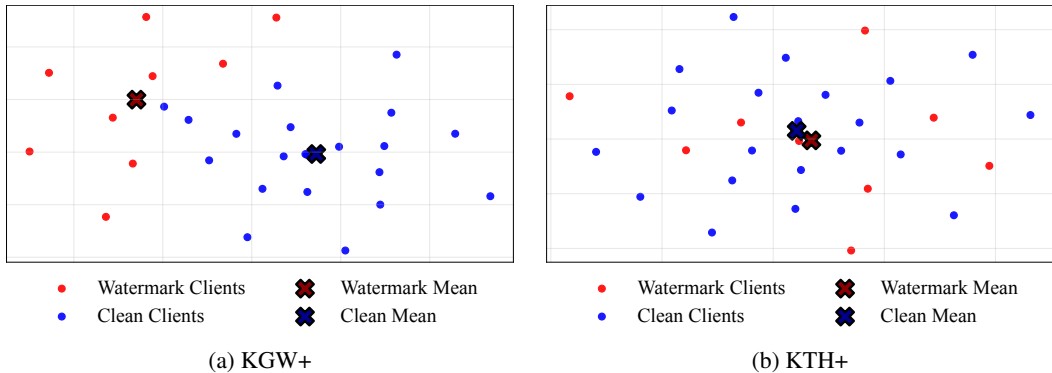

(a) KGW+          (b) KTH+

Figure 5: t-SNE visualizations of local updates from clean clients (blue) and watermarking clients (red) for KGW+ and KTH+ in the first round. The mean gradient for each group is marked with an 'X'. All clean clients use synthetic data.

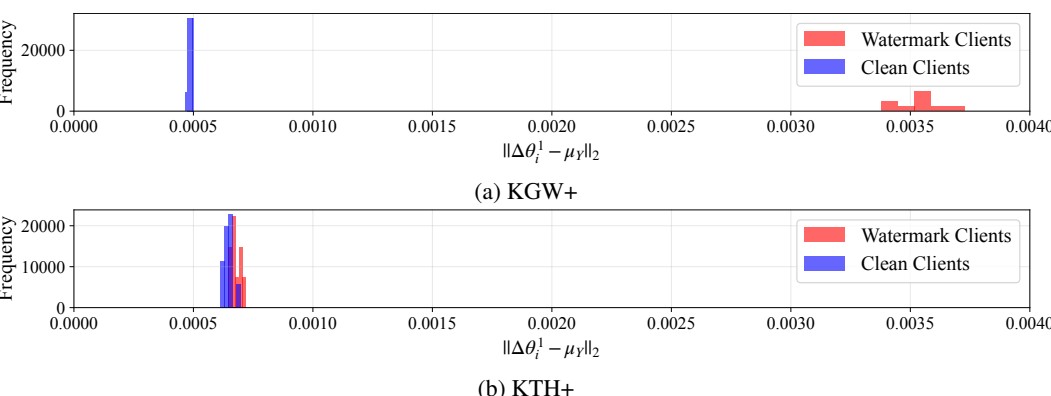

Figure 6: The distribution of the $\ell^2$-norm of displacement vector from each update to clean mean in the first round. All clean clients use synthetic data.

in practice. Under the same setup as the KGW+ experiments ($k$-gram = 4 and all clients fine-tune on synthetic data), Gumbel does not exhibit robustness (ER = 1.4%).

## G.2 NON-LLM WATERMARKS

There are non-LLM watermarking schemes applicable to the FL fine-tuning, such as post-hoc detection. We examine the Unicode character replacement scheme (Wei et al., 2024), which introduces a simple watermark by inserting random sequences or Unicode lookalikes into training data. This approach, independent of the LLM generation process, detects watermarks by analyzing token loss over these modified elements. We find that this watermark method lacks sufficient detection significance under the FL setting. We test the watermark on Pythia-70M with $\epsilon = 30\%$, leading to a $p$-value of $0.274$. In comparison, $p$-value for KGW+ under the same setting of $\epsilon = 30\%$ is $0.0041$. The likely reason for this insignificance is that the original method was designed for pre-training; FL fine-tuning appears inadequate for the watermark to become statistically discernible.

## H RELATED WORK

### H.1 LLM WATERMARKS

Zhao et al. (2025) identify two primary families of LLM watermarks. We explore one representative watermark from each.

**Green-Red Watermark.** Green-red watermark randomly partitions the vocabulary into "green" and "red" lists at each generation step and skews logits to favor green-listed tokens. Detection is performed via a statistical testing of the green-list token proportion in the generated text. Kirchenbauer et al. (2024) is one representative work in this family. It utilizes a hashing method to pseudo-randomly determine green list at each step, compared to approaches like Zhao et al. (2023), which predetermine a green list for all tokens. Subsequent research in this family focuses on addressing key limitations, including improving watermark robustness against adversarial attacks (Liu et al., 2024), enhancing the watermark performance in low-entropy scenarios (Lee et al., 2024), and increasing information capacity to carry multi-bit information (Wang et al., 2024a).

**Gumbel Watermark.** Kuditipudi et al. (2024) is a representative watermark from the Gumbel watermark family. It employs a secret random number sequence to manipulate token sampling while preserving the original output distribution. The watermark is statistically distortion-free and does not rely on hashing. Aaronson & Kirchner (2023) proposes an alternative approach, which is hashing-based and computationally distortion-free provided that the tokens used for hashing do not repeat (Zhao et al., 2025). We select Kuditipudi et al. (2024) over Aaronson & Kirchner (2023) because the latter's distortion-free guarantee depends on a precondition that may not hold in practice. Further analysis and experimental results for Aaronson & Kirchner (2023)'s approach are provided in Appendix G.

Sander et al. (2024) shows that hashing-based watermarks can exhibit *radioactive* properties in centralized setup. We adapt their detection method and investigate if the findings still hold in FL setup. We additionally include an active server threat model and show that it can effectively remove the watermarked updates.

## H.2 Federated Learning

**FedSGD vs. FedAvg.** Federated learning (FL) (McMahan et al., 2017) is a distributed machine learning framework where multiple clients collaboratively train a shared model under the coordination of a central server. Standard FL setting, FedSGD (McMahan et al., 2017), performs a single batch gradient over the entire local dataset per communication round. While straightforward, this approach typically requires a large number of communication rounds to converge, incurring high communication costs. FedAvg (McMahan et al., 2017) significantly reduces communication overhead by running multiple local epochs on clients before synchronizing with the server.

**FL Optimization.** FedSGD and FedAvg rely on SGD for both local client updates and server aggregation, which can lead to slow convergence, particularly in scenarios involving non-IID data distributions (Karimireddy et al., 2021) or environments with heavy-tailed random gradient noise distribution (Zhang et al., 2020). To address these challenges, many research explored the use of adaptive optimization methods to accelerate convergence. Several studies propose local adaptive optimization strategies, such as adaptively adjusting local learning rate (Xie et al., 2020; Sun et al., 2023), maintaining local momentum buffers (Yu et al., 2019), or applying adaptive local interval (Spiridonoff et al., 2020; Ji et al., 2020). FedOpt (Reddi et al., 2021) introduces adaptive server-side optimizers (e.g., ADAGRAD, ADAM, or YOGI), which we adopt as our FL framework due to its significant improvement in convergence speed. Specifically, in some of our experiments, while FedAvg requires over 200 rounds to converge, FedOpt achieves convergence in less than 50 rounds.

## H.3 Byzantine-robust Aggregation

FL systems that rely on a centralized server to simply average local client updates have been shown to be vulnerable to various adversarial attacks (Bagdasaryan et al., 2019; Bhagoji et al., 2019; Nasr et al., 2019; Sun et al., 2022). Even a small number of malicious clients can stealthily distort the global model through carefully crafted updates (Fang et al., 2020). Therefore, numerous Byzantine-robust FL protocols have been proposed to mitigate the impact of such malicious updates.

**Weak Robust Aggregators.** Weak robust aggregators (Blanchard et al., 2017; Chen et al., 2019; Yin et al., 2018) compute measures of centrality per dimension and provide weak theoretical guarantees on the maximum bias. Yin et al. (2018) propose aggregating local updates using coordinate-wise

trimmed mean, while others leverage geometric median (Chen et al., 2019) or Euclidean distances between vectors (Blanchard et al., 2017)). When applied to an $\epsilon$-corrupted set of $d$-dimensional vectors, none of these methods can achieve a total bias bound tighter than $O(\sqrt{\epsilon d})$ (Lai et al., 2016; Lugosi & Mendelson, 2021).

**Strong Robust Aggregators.** Strong robust aggregators compare magnitudes across all possible vector directions. They provides strong bias bounds independent of $d$ — a crucial security guarantee for high-dimensional ML models. Polynomial-time strong aggregators (Diakonikolas et al., 2018; Hopkins et al., 2021; Kothari & Steurer, 2017) identify outliers by examining their projections onto the dominant eigenvectors of the sample covariance matrix. While these polynomial-time methods offer strong theoretical guarantees, they often cannot achieve these bounds in practice due to computational limitations. Specifically, these methods require $O(\epsilon d^2)$ operations and $O(d^2)$ memory, rendering them impractical and vulnerable to attacks in high-dimensional settings (Choudhary et al., 2024; Lee et al., 2025).

We employ RandEigen (Lee et al., 2025) in our ActiveFL, which is a state-of-art strong Byzantine robust aggregator that runs in quasi-linear running time, $O(Nd)$, and possesses provably near-optimal bias bounds. Unlike polynomial-time strong aggregators, RandEigen does not require computing the maximum variance of clean vectors, $\|\Sigma_C\|_2$. Such design allows us to process an entire layer of the global model at once. In contrast, polynomial-time strong aggregators require splitting each layer into chunks of 1000 elements to meet practical memory constraints. Note that even with RandEigen, we must partition large layers (e.g. embedding layers of Pythia-70M) by factors of 8 or 16 to avoid memory overflow on an 80GB H100 GPU.

### H.4 Other Watermarking in FL

While concurrent work also explores client-side FL watermarking scheme, WAFFLE (Yang et al., 2023) adopts a backdoor-based approach tailored for image classification tasks. In contrast, our work focuses on data-based watermarks for NLP tasks, addressing unique challenges in text-based FL environments.

## I THE USE OF LARGE LANGUAGE MODELS

In preparing this manuscript, we made limited use of a large language model solely for the purpose of polishing the writing. Specifically, the LLM was used to improve grammar, clarity, and readability of sentences drafted by the authors. The LLM did not contribute to research ideation, experimental design, analysis, or the generation of novel content. All scientific contributions, methodology, and results reported in this paper are the sole responsibility of the authors.

