# OpenReview forum: "Watermark Robustness and Radioactivity May Be at Odds in Federated Learning"
_ICLR.cc/2026/Conference — ICLR 2026 Conference Withdrawn Submission_

### Official Review · Reviewer_Hc8f · 2025-10-26

**Soundness:** 2
**Presentation:** 2
**Contribution:** 2
**Rating:** 4
**Confidence:** 4

**Summary:**

The paper investigates the behavior of watermarking techniques in federated learning settings. The experiments reveal that watermark signals remain detectable when large language models are fine-tuned on datasets containing a small proportion of watermarked samples. However, an active adversarial server can identify watermarked synthetic samples as outliers and remove them to evade provenance tracking. The evaluation systematically explores these findings and addresses four key research questions.

**Strengths:**

+ The topic of watermarking in federated settings is interesting and important.

**Weaknesses:**

- The threat model could be more clearly defined. In particular, it remains unclear whether all users in the federated learning setting employ the same key and watermarking technique when synthesizing data samples.
- The main insight of the paper is that watermarking techniques introduce low distortion. However, more recent distortion-free watermarking methods have been proposed. The paper does not discuss how such techniques could be integrated into provenance tracking or how adversaries might evade them. It would also be valuable to analyze whether synthesized data from distortion-free versus low-distortion watermarking methods affect the model’s training performance differently.
- The evaluation considers only two watermarking techniques, which limits the generalizability of the conclusions. Including a broader range of watermarking methods and datasets would make the analysis more comprehensive and convincing.

**Questions:**

+ How does the adversarial server determine which clients have watermarked their datasets with synthetic samples and which have not? In particular, what criteria or detection methods are used to identify these outliers?
+ If a larger proportion, or even a majority, of clients employ watermarking to synthesize their data, how can the adversarial server still effectively detect or filter out watermarked samples under such conditions?

---

> ### Author Response · Authors · 2025-11-23
> **Rebuttal by Authors**
>
> We thank reviewer Hc8f for their time and valuable feedback. Below are our responses to the points raised.
>
> - **Threat model clarification (W1)**: Based on the reviewer’s feedback, we have conducted new experiments where clients employ different secret keys for watermark generation. Our core findings remain consistent: Even with this key heterogeneity, each KGW-watermarking worker (with a different key) demonstrates strong radioactivity in the vanilla setup, while KTH-watermarking workers do not (exhibiting high p-values ~0.5 after fine-tuning). The evasion rate of KGW watermarking workers is still as low as 2.36% in the adversarial setup. The table below demonstrates LLM watermarks radioactivity under vanilla setup.
>
>   **Table 1: KGW radioactivity under FL fine-tuning with Pythia-160M and C4**
>   | WM | p-val (worker 0) || p-val (worker 1) ||
>   |:----:|:-----------------------------:|:-----------------------------:|:-----------------------------:|:-----------------------------:|
>   | | **Before FT** | **After FT** | **Before FT** | **After FT** |
>   |KGW+| 0.355 | 7.44E-4 | 0.169 | 6.33E-6 |
>
> - **More distortion-free watermarking schemes (W2, W3)**: A key focus of our study is evaluating LLM watermark techniques under the FL setting. Therefore, we picked representative watermarks from both the low-distortion and distortion-free family. Specifically, we found that the KTH+ watermark, despite being statistically distortion-free, is not radioactive under FL. In response to reviewer feedback, we expanded our analysis to include one more computationally distortion-free watermark: AAR. The tension between radioactivity and robustness remains consistent. While AAR is radioactive in the vanilla setup, it is filtered out under adversarial setup, with evasion rates of 1.94% for 4-gram and 0.67% for 10-gram. Table 2 shows AAR radioactivity under vanilla setup, and Table 3 shows AAR robustness under adversarial setup, demonstrating that AAR is also ineffective under such FL setup.
>
>   **Table 2: AAR radioactivity under FL fine-tuning with Pythia-160M and C4**
>   | WM | k-gram | p-val (worker 0) | | p-val (worker 1) | |
>   |:-----:|:-----:|:-----:|:-----:|:-----:|:-----:|
>   | | | **Before FT** | **After FT** | **Before FT** | **After FT** |
>   | AAR | 4 | 0.923 | 1.46E-8 | 0.245 | 8.82E-9 |
>   | AAR | 10 | 0.711 | 2.74E-5 | 0.776 | 1.85E-5 |
>
>   **Table 3: AAR robustness under FL fine-tuning with Pythia-160M and C4**
>   | WM | k-gram | Evasion Rate | Overfiltering Rate |
>   |:--:|:----:|:-------:|:-------:|
>   | AAR | 4 | 1.94% | 53.55% |
>   | AAR | 10 | 0.67% | 48.0% |
>
> - **Clarification (Q1)**: We do not assume that the server knows which clients have watermarked gradients. Instead, the server estimates and filters the watermarked gradients out using a robust Byzantine-tolerant aggregation scheme. Specific criteria are presented in Section 4 and Appendix C.
>
> - **Varied watermark ratio (Q2)**: There are statistical limitations to the watermark ratio that the strong Byzantine-robust aggregator can filter. Although the aggregator's theoretical guarantee is limited to an 8.3% outlier ratio, our experiments demonstrate its practical effectiveness extends to ratios of up to 30%.

---

### Official Review · Reviewer_j6vk · 2025-10-29

**Soundness:** 2
**Presentation:** 3
**Contribution:** 1
**Rating:** 2
**Confidence:** 4

**Summary:**

This paper studies the use of LLM watermark radioactivity in federated learning to enable clients to determine whether the model was trained on their data. More specifically, the authors show that a malicious server could distinguish updates from clients using watermarked text and non-watermarked text, thereby evading accountability.

**Strengths:**

- This is the first work that studies whether the watermark radioactivity properties still hold in the federated learning setting, where multiple gradients from watermarked and unwatermarked data are merged.
- The method is well explained from an intuitive perspective.

**Weaknesses:**

- The actors' goals are hard to understand. In l.135, the authors write `In FL provenance, an ε fraction of clients, that we denote as watermarking clients, aim to prove that their datasets were used to train the global model.` This raises the questions:
	- If the clients want to know whether their data are used to train the model, why do they send their updates in the first place?
	- Because the clients send their updates, do they not already know that their data are used to train the model?
	- Even if they use the watermark and the server is malicious, they still know whether their data were used to train the model:
		- If the p-value is high, it is because the malicious server successfully removed the watermarked updates, thus the data were NOT used to update the model. This agrees with the p-value result.
		- If the p-value is low, the malicious server was not successful in removing the watermarked updates, thus the data were used to update the model. This also agrees with the p-value result.
	- Lastly, because the clients know the weights of the model, they can easily identify which model in the wild was trained with their data by using a provenance method relying on the weights.
- The evaluation is not complete enough:
	- Utility evaluation:
		- The current evaluation only evaluates the impact of watermarked updates on the model entropy, not the impact of the method per say. If the (watermarked) updates are poor with respect to quality, this is independent of the method.
		- What would be interesting is the impact on utility when using the robust aggregator with only clean data. Moreover, if the data come from non-IID clients, can you lose some impactful capabilities because of the robust aggregation?
		- For measuring utility, using LLM benchmarks (HumanEval, GSM8k, MMLU, ...) would be more meaningful than the entropy of the model.
	- The models used are both small and old. The authors noticed that the larger the model, the higher the radioactivity. What would happen with recent 3B or 8B models? Would the small evasion rate be enough for those models to be radioactive?
	- Using i.i.d. clients is not realistic. I think the method works because there are two distributions among the clients, the watermarked and the non-watermarked. What if each client has its own domain-specific dataset that is either human or synthetic? In this case, would the method easily separate watermarked text?
	- The authors only use two watermarking schemes, and only one is successful. It is unclear whether the attack still works against more advanced watermark schemes.
- The novelty is limited: the malicious server method is standard outlier detection, and watermark robustness is already well-studied (though not in the FL setting). In particular, the authors claim in l.74 that they adapt existing watermarking schemes, but this is not true; the schemes are used as is.
- The definition of robustness is not useful: without putting constraints on the adversary's capabilities, the existence of the adversary is almost certain (an oracle adversary could know the watermark updates with 100% accuracy).

**Questions:**

- Could the authors explain what is the goal of the different actors, i.e. what are they trying to accomplish and why?
- Could they evaluate the impact of their methods on utility when no watermarked updates are present, especially using LLM benchmarks and non-iid data? (see the weakness section for more details)
- Could the authors show whether the malicious server still succeeds with bigger and newer models?
- What happens if the clients are not iid? Is the method still able to distinguish watermarked updates? Is the utility impacted when no watermarked clients are present because "useful" updates are filtered?
- Could you explain why you dont use distortion-free hashing based schemes like AAR or SynthID-text? Especially given that you acknowledge in l340 that they are radioactive. Could the fact that they are distortion-free bypass the outlier detection method when all clients are synthetic data?
- Can you justify the claims that KTH detection can not aggregate other the prompts (l341 and 353)? Why can't you just concatenate the different prompts given that, as mentioned in l783, `If none of the clients apply watermarking, the data is independent and identically distributed.`?
- OFR is quite high (around 50%). How is this not an issue for utility/training? Especially when no clients are watermarked, OFR is at above 90%.

---

> ### Author Response · Authors · 2025-11-23
> **Rebuttal by Authors**
>
> We thank reviewer j6vk for their time and valuable feedback. Below are our responses to the points raised, and we included relevant references in the general response.
> - **Motivation (W1, Q1)**: Prior FL watermarking work [9] focuses on model ownership, allowing the server to prove IP ownership. However, it does not enable individual clients to verify if their data contribute to the global model, which is particularly problematic when there is unknown downstream model deployment. Recent evidence outside FL shows that models are frequently released or repurposed in ways that may contradict the terms of the original training data, such as being adapted for military application [10], creating competing derivative works that harm the original data's economic value [11, 12], or being sold for profit by other malicious clients [9]. Watermarking clients’ goal is thus to audit if a downstream model in the wild was trained on their data. In this realistic scenario, clients only have black-box query access to such suspect models.
>   The clients know that their watermark is effective, if the p-value is low. However, a high p-value is ambiguous under malicious servers. It could mean that the watermark has is not *yet* radioactive or the server removed the watermark. Note that our adversary does not discard a watermarking client's full update but selectively filters a subset of layers that are outliers, i.e., the high p-value does not mean that the client’s data was not used.
> - **Evaluation metrics (W2, Q4)**: We reported more utility metrics in Table 9 (Appendix E.4), including word-level perplexity (C4) and Q&A accuracy (MMLU-Pro). We focus on mixed synthetic and natural data in this paper, consistent with prior work [1].
> - **All clean data (Q2, Q4, Q7)**: When all clients fine-tune with clean data, most strong robust aggregators correctly retain all client gradients because such gradient mean naturally aligns with an estimated clean mean. RandEigen can over-filter initially as it depends on the changes in the largest eigenvalue variance. This can be mitigated by a more relaxed threshold in the first few rounds.
> - **Non-IID (Q5)**: We use similar experimental setups as other FL LLM fine-tuning and aggregation works [3, 4]. We agree that this is still a subset of the complexities of real-life FL.
> - **More LLMs (W2, Q3)**: Due to computational constraints, we were unable to evaluate larger models.
> - **Novelty (W3)**: Existing watermark attacks [5, 6, 7] operate on text, e.g., via paraphrasing or translation. However, in FL only the model updates are available, preventing the server from directly determining which clients watermark. While low-distortion watermarks introduce bias in the text space, this bias does not necessarily translate into algorithmically detectable deviation in the parameter space. Practical robust aggregators tolerate bias proportional to the model parameter size $d$, i.e., $\Omega(\sqrt{\epsilon d})$ for $\epsilon < 50$%. Given that $d$ is very large ($>10^7$) for LLM, the allowable bias can be substantial. Our work, therefore, re-evaluates watermark robustness in the FL setup due to two critical distinctions: 1) the server's inability to directly inspect client text, and 2) the potential for watermark-induced bias to remain within the tolerance of robust aggregation.
> - **Robustness (W4)**: The adversary is computationally efficient and maintains the model's utility relative to the vanilla setup (Definition 2, Section 3). We will clarify it in the paper.
> - **Distortion-free watermarking schemes, AAR (W2, Q5)**: We evaluated the computationally distortion-free watermarking scheme, AAR. While AAR is radioactive in the vanilla setup, it is easily filtered out under adversarial setup, confirming our paper’s findings. This vulnerability arises because the scheme's distortion-free property depends on non-repeating k-grams, an assumption commonly violated by the repetitive nature of LLM generation. Table 1 shows AAR radioactivity under vanilla setup, and Table 2 shows AAR robustness under adversarial setup, demonstrating that AAR is similarly ineffective in FL.
>
>   **Table 1: AAR radioactivity under FL fine-tuning with Pythia-160M and C4**
>   | WM | k-gram | p-val (worker 0) | | p-val (worker 1) | |
>   |:-----:|:-----:|:-----:|:-----:|:-----:|:-----:|
>   | | | **Before FT** | **After FT** | **Before FT** | **After FT** |
>   | AAR | 4 | 0.923 | 1.46E-8 | 0.245 | 8.82E-9 |
>   | AAR | 10 | 0.711 | 2.74E-5 | 0.776 | 1.85E-5 |
>
>   **Table 2: AAR robustness under FL fine-tuning with Pythia-160M and C4**
>   | WM | k-gram | Evasion Rate | Overfiltering Rate |
>   |:--:|:----:|:-------:|:-------:|
>   | AAR | 4 | 1.94% | 53.55% |
>   | AAR | 10 | 0.67% | 48.0% |
>
> - **KTH detection (Q6)**: We thank the reviewer for their insightful suggestion. We evaluated the KTH+ watermark with concatenated prompts in the vanilla setup. The results confirm that **the watermark is non-radioactive**, with a high p-value ~0.5 after fine-tuning.

---

> > ### Comment · Reviewer_j6vk · 2025-11-24
> >
> > I thank the authors for their reply.
> >
> > **Motivation** This is a major change to the paper that should be reflected in the next submission. Given the clarified threat model, I have some additional concerns:
> > - If the goal is to study downstream adaptation of the model, would not the issues from [1] (i.e. the watermark radioactivity fades out after downstream modification to the model) limit the usability of the method.
> >
> > **Evaluation Metric, clean data and non-iid** Given that the technical contributions of this work are limited (Byzantine-robust aggregation and watermarking are existing out-of-the-box techniques), the major contributions are (i) to highlight a new underexplored threat model and (ii) to show how to mitigate it in practice. The second aspect requires extensive evaluation with realistic scenarios. Using non-IID clients is critical for this point.
> >
> > **KTH experiment** Where do the authors include the new KTH experiment? Their replies lack the technical details (experimental setup) needed for me to fully understand what they did. In particular, I am surprised by their results, as [2] shows that, with SFT, KTH is radioactive. Is this change in behavior induced by the more complicated FL setup?
> >
> > **Newer/Larger models** The authors report using an H100 GPU for their experiments. I believe 1B/3B-parameter models should fit on such a GPU. Note that this is nonetheless a minor point.
> >
> >
> >
> > [1] Towards Watermarking of Open-Source LLMs, Gloaguen et al., arXiv 2025\
> > [2] On the Learnability of Watermarks for Language Models, Gu et al., ICLR 2024

---

> ### Author Response · Authors · 2025-12-01
>
> We thank the reviewer again.
>
> **Motivation.** We have not changed the threat model, and only clarified why data provenance is useful. The problem of whether watermarks are robust to downstream modifications (presented in [1]) is orthogonal to this paper. Our contribution is to show 1) which watermarks are radioactive in federated learning (similar to [2] but in the FL case) and 2) present a new threat model when an active server filters the updates.
>
> **KTH experiment.** The primary reason for having different results from [2] is the choice of sampling method. [2] employed exponential minimum sampling (EXP-EDIT), while we tested with inverse transform sampling (ITS). These two different watermarking schemes are both proposed in the KTH paper [3]. The KTH authors state that EXP-EDIT has better detection power, and the mechanism is very similar to AAR; therefore, it is not surprising that KTH (EXP-EDIT) demonstrates radioactivity under centralized setting. We tested EXP-EDIT under a similar setting as [2], utilizing the same predefined key but varying offset across the prompts. The results show that such EXP-EDIT setting is not robust to aggregation, as the evasion rate $ER = 11$\%. Due to limited time, we did not have the chance to run EXP-EDIT using different keys across the prompts. We plan to conduct these experiments and provide a more comprehensive discussion in the revised version of our paper.
>
> [1] Towards Watermarking of Open-Source LLMs, Gloaguen et al., arXiv 2025
>
> [2] On the Learnability of Watermarks for Language Models, Gu et al., ICLR 2024
>
> [3] Rohith Kuditipudi, John Thickstun, Tatsunori Hashimoto, and Percy Liang. Robust distortion-free watermarks for language models, 2024. URL https://arxiv.org/abs/2307.15593.

---

### Official Review · Reviewer_WjYY · 2025-10-31

**Soundness:** 2
**Presentation:** 3
**Contribution:** 2
**Rating:** 2
**Confidence:** 4

**Summary:**

This work investigates watermark radioactivity in federated learning. The authors first demonstrate that watermarked data exhibits radioactive behavior in federated learning systems. They then formulate an active adversary threat model, in which a malicious server attempts to identify model updates originating from watermarking clients and exclude them from aggregation. Several experiments are conducted, showing that existing watermarking methods fail to evade detection by robust aggregation mechanisms.

**Strengths:**

1. This work addresses the problem of data provenance, which is a critical and timely topic.

2. The authors identify the potential existence of an active adversary aiming to maintain model utility while evading provenance tracking, highlighting an important and realistic threat scenario.

3. Experimental results demonstrate that none of the existing watermarking methods simultaneously achieves radioactivity, robustness, and utility, providing valuable insights into the trade-offs and challenges of watermarking in federated learning.

**Weaknesses:**

1. Beyond formulating the problem of how an adversary can evade provenance tracking, this work lacks substantial technical contributions. Both the watermarking methods and the robust aggregation mechanisms are directly borrowed from prior works.

2. The motivation for federated learning provenance is not well established. Specifically, as stated in lines 134–135, “In FL provenance, an $\epsilon$ fraction of clients, denoted as watermarking clients, aim to prove that their datasets were used to train the global model.” However, this assumption seems questionable—clients in an FL system already agree to contribute their data for model training, so it is unclear why they would later need to prove their participation. Clarifying this motivation is essential to justify the relevance and practicality of the problem formulation.

3. The experimental evaluation is limited in scope, particularly regarding the diversity and scale of LLMs. More widely adopted and representative models, such as LLaMA and Gemma, are not included in the evaluation. Additionally, the models used in the experiments are relatively small; incorporating larger and more realistic models would provide stronger and more generalizable results.

**Questions:**

1. In lines 43–45, the authors state that “models fine-tuned on watermarked LLM-generated text exhibit radioactivity in a centralized setting, where watermark signals remain detectable after fine-tuning,” citing Sablayrolles et al. (2020) and Sander et al. (2024). However, in lines 51–53, they claim that “continued fine-tuning on non-watermarked (clean) data can substantially reduce watermark detectability,” citing Sander et al. (2024) again. These two statements are contradictory, as one suggests watermark persistence while the other implies its removal, both referencing the same work. Why does the author make these two contradictory statements in the paper?


2. Many important details about the fine-tuning process on local clients are missing. It is unclear whether clients fine-tune their models using PEFT techniques (e.g., LoRA) or full model fine-tuning. Such details are critical for reproducibility and for understanding the impact of local training on watermark detectability.


3. Figure 1 shows that updates from clean and watermarked clients can be clearly separated in the feature space. This raises an important question: why is a strong Byzantine aggregator still necessary if the two groups are already distinguishable? Furthermore, the visualization suggests a potential for designing a new filtering mechanism. For instance, if the updates naturally form two clusters and clean clients constitute the majority (as assumed in the paper), the smaller cluster could be interpreted as the watermarking clients and filtered out directly.

---

> ### Author Response · Authors · 2025-11-23
> **Rebuttal by Authors**
>
> We thank reviewer WjYY for their time and valuable feedback. Below are our responses to the points raised, and we included relevant references in the general response.
>
> - **Technical contributions (W1)**: We agree that designing a watermark that is robust in this setup is not presented in this paper; however, we think that it is a promising direction for future work and we will consider these insights in our subsequent research.
>
> - **Motivation (W2)**: Prior FL watermarking work [9] focuses on model ownership, allowing the server to prove IP ownership. However, it does not enable individual clients to verify if their data contribute to the global model, which is particularly problematic when there is unknown downstream model deployment. Recent evidence outside FL shows that models are frequently released or repurposed in ways that may contradict the terms of the original training data, such as being adapted for military application [10], creating competing derivative works that harm the original data's economic value [11, 12], or being sold for profit by other malicious clients [9]. This is why data provenance is crucial. Watermarking clients’ goal is thus to audit whether a downstream model in the wild was trained on their data. In this realistic scenario, clients only have black-box query access to such suspect models.
>
>   The clients know that their watermark is effective, if the p-value is low. Our observation is that a high p-value is ambiguous under malicious servers. It could mean that the watermark has not *yet* become radioactive or the server removed the watermark signal. Note that our adversary does not discard a watermarking client's full update but selectively filters a subset of layers that behave as outliers, i.e., the high p-value does not mean that the client’s data was not used.
>
> - **More LLMs (W3)**: We agree that employing larger and more recent LLMs would provide stronger and more generalizable results. Unfortunately, our study was limited by computational constraints as larger models require extended fine-tuning time to converge. We believe our findings establish a foundational insight that will generalize, and we plan to validate this with more powerful models in future work.
>
> - **Clarifications (Q1, Q2)**: We thank the reviewer for their careful reading. Regarding question 1, the two statements are not contradictory, and we include them for different purposes to describe a sequence of findings from Sander et al. [1]:
>    - Line 43–45: A model fine-tuned on watermarked text becomes radioactive .
>    - Line 51–53: Subsequently fine-tuning this radioactive model on clean data weakens the watermark signal but does not erase it, as the model remains statistically detectable.
> We will revise the text to clarify.
>
>   For question 2, we did full model fine-tuning for every client local training. We thank the reviewer for pointing out the missing detail. We will add this to the paper.
>
> - **Simpler filtering algorithm (Q3)**: We employ a robust aggregation method instead of simple client-level filtering to avoid discarding the entire updates from watermarking clients or discarding updates in an ad-hoc manner which might degrade the model utility. Specifically, the aggregator filters on the basis of layers. For example, even with a KGW+ watermark, updates to the embedding layer are often preserved. This **enables learning from watermarked clients while filtering** out highly deviated gradients that lead to high radioactivity.

---

### Official Review · Reviewer_fdMF · 2025-11-01

**Soundness:** 2
**Presentation:** 3
**Contribution:** 1
**Rating:** 2
**Confidence:** 4

**Summary:**

The paper explores whether data provenance in federated learning can be achieved through watermarking text produced by large language models. It studies two statistical watermarking methods, KGW+ and KTH+, in a setting where multiple clients collaboratively fine-tune a shared model. When the server simply averages the client updates, the watermark signal persists and the resulting global model still reflects traces of the marked data. When the server instead applies a robust aggregation method that filters unusual updates, the watermark signal disappears while the model's accuracy remains stable. The authors interpret this behavior as evidence of a tension in federated training: model utility, robustness, and detectable provenance cannot all be achieved simultaneously.

**Strengths:**

The study is thorough in its empirical design. It tests two watermarking schemes across several model sizes in the Pythia family and varies the proportion of watermarked data from small to moderate levels. The main finding is clean: when the server aggregates client updates with a robust method, the watermark signal largely disappears while model performance remains unchanged. This provides a clear picture of how robustness in federated learning can suppress provenance signals that rely on small statistical traces.

**Weaknesses:**

1. The paper sets out to test whether watermarking can serve as a foundation for data provenance in federated learning, yet the experiments ultimately show that the idea fails once the setting becomes realistic. The limitation lies in the premise more than in the execution. Statistical watermarks are already known to be fragile: their signal weakens with continued training and can be removed with little effort, as demonstrated by Zhang et al. (2023), "Watermarks in the Sand...". In this light, the finding that federated aggregation erases the watermark confirms a known vulnerability rather than uncovering a new phenomenon. A stronger study would begin with a clear argument for why watermarking might succeed in such an environment before investigating its breakdown.

2. The analysis considers only statistical watermarking methods that introduce small sampling biases into the model's output. As the share of watermarked data grows, these biases distort learning and reduce model performance. The authors interpret this behavior as evidence of an inherent trade-off between watermarking and utility, but the effect arises from the particular design of the watermark rather than from any general incompatibility. Cryptographic watermarking approaches based on pseudo-random codes (Christ and Gunn, 2024) define a better design space. Such schemes embed information through computational indistinguishability rather than statistical bias, and therefore would not degrade utility in the same way. Because this line of work is not discussed, the paper's conclusions remain narrower than they appear.

3. The experiments assume that a subset of clients share the same watermark key and the same generative model. This simplifies the analysis but does not reflect realistic federated systems, where clients hold distinct data, use different local processes, and keep their parameters private from the server. The separability observed in the experiments arises from this synchronization rather than from any intrinsic property of federated learning. The assumption also explains why using LLM watermarking for provenance in FL is a bad idea. Provenance requires independence among clients and limited trust in the server, whereas watermarking depends on shared structure and coordinated keys.

4. The central finding, that robust aggregation removes watermark signals while preserving accuracy, is empirically clear but theoretically unsurprising. Robust aggregation is designed to filter outliers, and gradients shaped by watermarking behave like outliers. The paper reports this effect but stops short of analyzing why it occurs or whether a different watermarking design might resist it. The analysis also ends at thirty percent watermarking clients, leaving open what happens at higher or lower proportions and why performance degrades as the watermark share increases. A closer examination of these dynamics, perhaps contrasting statistical and pseudo-random code watermarking, would make the contribution more substantial.

**Questions:**

1. Why was the study limited to detectable statistical watermarks? How would cryptographically undetectable (PRC-style) schemes behave under the same setup?
2. Why are all watermarking clients assumed to share the same key and generative model? How would heterogeneity in keys or generators affect your results?
3. Have you tested beyond \epsilon = 30% to confirm whether the reported utility–robustness trade-off persists or saturates?
4. What practical insight should readers take from this result, given that statistical watermarks are already known to be non-robust outside federated learning?

---

> ### Author Response · Authors · 2025-11-23
> **Rebuttal by Authors**
>
> We thank reviewer fdMF for their time and valuable feedback. Below are our responses to the points raised, and we included relevant references in the general response.
>
> - **Technical novelty (W1, W4, Q4)**: Our work starts with KGW+ watermarks that are highly radioactive in the centralized setting. We find that KGW+ is highly radioactive using **standard federated aggregation** (the vanilla-FL case, see Table 1, Section 5). Existing watermark removal attacks [5, 6, 7] operate by directly modifying text, e.g., via paraphrasing or translation. However, in FL only the model updates are available, preventing the server from directly determining which clients embed the watermark. [1] considers a "purification" approach which only requires more clean data as a watermark removal mechanism but finds it ineffective. Thus, our attack is different in two main ways: 1) It shows that watermarks are detectable in the model parameter space and 2) Only a subset of the watermarking updates are removed, making it a selective removal attack based on the outlier information.
>
>
>      While low-distortion watermarks introduce bias in the text space, this bias does not necessarily translate into algorithmically detectable deviation in the parameter space. Practical robust aggregation algorithms tolerate bias proportional to the model parameter size $d$, i.e., $\Omega(\sqrt{\epsilon d})$ for $\epsilon < 50$%. Randeigen is $d$ independent but up to $\epsilon< 8.3$%. Thus, it was not clear that even radioactive watermarks such as KGW+ would be filtered out at higher $\epsilon$, or that distortion-free watermarks are not radioactive.
>
> - **Distortion-free watermarking schemes, PRCs (W2, Q1)**: We also evaluated KTH, which is a **statistically distortion-free** watermarking method. While the reviewer points to PRC, this watermark has not yet been implemented as a practical LLM watermarking solution even in centralized settings [2]. However, we expanded our analysis to **include one computationally distortion-free** watermark: AAR [8]. The tension between radioactivity and robustness remains consistent. While AAR is radioactive in the vanilla setup, it is filtered out under adversarial setup, with evasion rates of 1.94% for 4-gram and 0.67% for 10-gram. Table 1 shows AAR radioactivity under vanilla setup, and Table 2 shows AAR robustness under adversarial setup, demonstrating that AAR is also ineffective.
>
>   **Table 1: AAR radioactivity under FL fine-tuning with Pythia-160M and C4**
>   | WM | k-gram | p-val (worker 0) | | p-val (worker 1) | |
>   |:------:|:------:|:------:|:------:|:------:|:------:|
>   | | | **Before FT** | **After FT** | **Before FT** | **After FT** |
>   | AAR | 4 | 0.923 | 1.46E-8 | 0.245 | 8.82E-9 |
>   | AAR | 10 | 0.711 | 2.74E-5 | 0.776 | 1.85E-5 |
>
>
>    **Table 2: AAR robustness under FL fine-tuning with Pythia-160M and C4**
>    | WM | k-gram | Evasion Rate | Overfiltering Rate |
>    |:--:|:----:|:--------:|:--------:|
>    | AAR | 4 | 1.94% | 53.55% |
>    | AAR | 10 | 0.67% | 48.0% |
>
> - **FL assumptions (W3, Q2)**: Based on the reviewer’s feedback, we added new experiments where clients employ different secret keys for watermark generation. Our core findings remain consistent: Even without key coordination, each KGW-watermarking worker demonstrates strong radioactivity on a per-worker basis in the vanilla setup. The results for Pythia-160M and dataset C4 are provided in the attached table. We use similar experimental setups as other FL LLM fine-tuning and aggregation works such as [3, 4]. We agree that this is still a subset of the complexities of real-life federated setups.
>
>   **Table 3: KGW radioactivity under FL fine-tuning with Pythia-160M and C4**
>   | WM | p-val (worker 0) || p-val (worker 1) ||
>   |:----:|:-----------------------------:|:-----------------------------:|:-----------------------------:|:-----------------------------:|
>   | | **Before FT** | **After FT** | **Before FT** | **After FT** |
>   |KGW+| 0.355 | 7.44E-4 | 0.169 | 6.33E-6 |
>
>
> - **Varied watermark ratio (W4, Q3)**: We evaluated the impact of varied watermark ratios (0% to 30%) on model utility using word-level perplexity (C4) and QA accuracy (MMLU-Pro), see Table 9, Appendix E.4. In summary, increasing the watermark ratio degrades performance. We observe utility fluctuation around a 30% watermark ratio, but we did not test beyond this. We hypothesize that the overall degradation stems from either the distributional shift induced by the watermarking or the synthetic nature of the data itself.
>
> - **Practical insights (W4, Q4)**: We agree that designing a watermark that is robust in this setup is a promising direction, which we leave for future work. Our paper’s contribution is to introduce data provenance via watermarking in this new problem space and characterize its vulnerability. We believe our work points to new watermarking designs that have to ensure more alignment with the semantics of the non-watermarked distribution in latent space.

---

### Author Response · Authors · 2025-11-23
**General Response by Authors**

# General Response

We thank all reviewers for their insightful comments and constructive critiques. We will revise the paper to address all the points raised. We first provide a high-level general response here.

Our primary goal was to propose and investigate the viability of existing watermarking schemes for data provenance in federated learning (FL). We demonstrated that hashing-based statistical watermarks successfully exhibit radioactivity under simple federated aggregation, yet these signals can be easily eliminated by Byzantine-robust aggregation. Conversely, distortion-free watermarks that avoid hashing show no radioactivity in our evaluated settings. We conclude that the existing LLM watermarks cannot reliably ensure data provenance in FL, and our findings provide critical insights for future watermark design in the FL setup.

- **Other watermarking schemes (fdMF, j6vk, Hc8f)**: We added evaluation for one more distortion-free scheme: AAR [8]. It confirms our findings for KGW+ and KTH+, and we have included this data. We note that **distortion-free schemes remain robust**, which can be shown theoretically under the assumption that in the latent space the text watermarks remain distortion-free. However, the evaluated schemes **are not radioactive**. We will extend our evaluation for this in the final version.

- **Motivation (j6vk, WjYY)**: We thank the reviewers for their insightful comments regarding the paper's motivation. We will revise the paper to better articulate the problem we address: enabling data provenance in FL to detect downstream model misuse. Addressing this problem is critical, as recent evidence shows that models are frequently released or repurposed in ways that may contradict the terms of the original training data, such as being adapted for military application [10], creating competing derivative works that harm the original data's economic value [11, 12], or being sold for profit by other malicious clients [9].

- **Setup variants (fdMF, j6vk)**: We recognize that the problem space is extensive, with many potential variations. A full exploration of all its variations such as those highlighted by the reviewers concerning alternative watermarking ratios and larger LLMs is beyond the scope of this work but remains valuable and important directions for future investigation. We believe our work provides a solid starting point for future work.

**References**

[1] Tom Sander, Pierre Fernandez, Alain Durmus, Matthijs Douze, and Teddy Furon. Watermarking makes language models radioactive, 2024. URL https://arxiv.org/abs/2402.14904.

[2] Xuandong Zhao, Sam Gunn, Miranda Christ, Jaiden Fairoze, Andres Fabrega, Nicholas Carlini, Sanjam Garg, Sanghyun Hong, Milad Nasr, Florian Tramer, Somesh Jha, Lei Li, Yu-Xiang Wang, and Dawn Song. Sok: Watermarking for ai-generated content, 2025. URL https://arxiv.org/abs/2411.18479.

[3] Sarthak Choudhary, Aashish Kolluri, and Prateek Saxena. Attacking byzantine robust aggregation in high dimensions, 2024. URL https://arxiv.org/abs/2312.14461.

[4] De Zhang Lee, Aashish Kolluri, Prateek Saxena, and Ee-Chien Chang. A practical and secure byzantine robust aggregator, 2025. URL https://arxiv.org/abs/2506.23183.

[5] Hanlin Zhang, Benjamin L Edelman, Danilo Francati, Daniele Venturi, Giuseppe Ateniese, and Boaz Barak. Watermarks in the sand: Impossibility of strong watermarking for generative models. arXiv preprint arXiv:2311.04378, 2023.

[6] Kalpesh Krishna, Yixiao Song, Marzena Karpinska, John Wieting, and Mohit Iyyer. Paraphrasing evades detectors of ai-generated text, but retrieval is an effective defense, 2023. URL https://arxiv.org/abs/2303.13408.

[7] Rohith Kuditipudi, John Thickstun, Tatsunori Hashimoto, and Percy Liang. Robust distortion-free watermarks for language models, 2024. URL https://arxiv.org/abs/2307.15593.

[8] Scott Aaronson and Hendrik Kirchner. Watermarking GPT outputs. https://scottaaronson.blog/?m=202302, 2023.

[9] Yuxin Yang, Qiang Li, Yuan Hong, and Binghui Wang. Fedgmark: Certifiably robust watermarking for federated graph learning, 2024. URL https://arxiv.org/abs/2410.17533.

[10] Ziad Assaad. Meta now allows military agencies to access its AI software. It poses a moral dilemma for everybody who uses it, 2024. URL https://doi.org/10.64628/aa.ydqkr4hhs.

[11] Shayne Longpre, Robert Mahari, Naana Obeng-Marnu, William Brannon, Tobin South, Katy Gero, Sandy Pentland, and Jad Kabbara. Data authenticity, consent, provenance for ai are all broken: what will it take to fix them?, 2024. URL https://arxiv.org/abs/2404.12691.

[12] Shayne Longpre, Robert Mahari, Anthony Chen, Naana Obeng-Marnu, Damien Sileo, William Brannon, Niklas Muennighoff, Nathan Khazam, Jad Kabbara, Kartik Perisetla, Xinyi Wu, Enrico Shippole, Kurt Bollacker, Tongshuang Wu, Luis Villa, Sandy Pentland, and Sara Hooker. The data provenance initiative: A large scale audit of dataset licensing attribution in ai, 2023. URL https://arxiv.org/abs/2310.16787.

---

### Author Response · Authors · 2025-12-01
**Withdrawal Decision**

We thank the reviewers for their time and feedback. We are withdrawing our paper to improve for the next submission.

---

### Note · Authors · 2025-12-01

I have read and agree with the venue's withdrawal policy on behalf of myself and my co-authors.